# Stable Spectral Copula Alignment for Robust Multimodal Learning

**Hongkang Zhang** [1]  **Shao-Lun Huang** [1]  **Yanlong Wang** [1]  **Ercan Engin KURUOGLU** [1]

## Abstract

Multimodal alignment can fail under deployment shift because standard objectives entangle cross-modal dependence with marginal-sensitive geometry. Stable Spectral Copula Alignment (SSCA) provides a deployment protocol for copula-stable dependence under approximately coordinate-wise monotone marginal distortions, together with auditable, label-free diagnostics for monitoring and mitigation. SSCA combines (i) clipped soft-rank Gaussianization that suppresses marginal effects while tracking tie and approximation errors, (ii) dependence-weighted sliced Wasserstein hub coupling for globally coherent multiway alignment with cycle auditing, and (iii) diagonal-stabilized block-spectral learning with eigengap-normalized Davis–Kahan diagnostics, yielding an actionable subspace-risk inequality. A calibrated gate maps diagnostic proxies to a reliability signal with a measurable false-alarm/miss trade-off, enabling stability-mode updates, budgeted remediation, and conservative no-update fallback for out-of-scope drift. Evaluations on MOSEI/MELD, MSCOCO, and CC3M-500K show improved performance under perturbation and substantially reduced degradation under controlled monotone distortions, raw-pipeline drifts, and frozen-feature retrieval stress tests.

## 1. Introduction

Multimodal alignment systems can fail under deployment shift because many objectives entangle cross-modal dependence with marginal-sensitive geometry. Monotone or approximately monotone marginal distortions in feature space, such as gain changes, normalization drift, and moderate quantization, can silently corrupt alignment even when cross-modal dependence remains informative. Raw-pipeline

changes such as compression and tokenizer changes are treated as empirical stress tests and remain outside the formal invariance claim. SSCA targets copula-invariant dependence via rank-based Gaussianization, multiway hub coupling, and diagonal-stabilized spectral learning, while exposing auditable diagnostics that support calibrated gating, budgeted remediation, and conservative fallback.

Copula theory (Sklar, 1959; Nelsen, 2006; Joe, 2014) decouples cross-modal dependence from marginal distributions: the copula is invariant under strictly increasing marginal transforms. Rank-based dependence summaries, including Spearman's $\rho$ and Kendall's $\tau$ (Spearman, 1904; Kendall, 1938), provide additional monotone-invariant signals that complement copula-based diagnostics. SSCA therefore uses a declared stability contract: when deployment drift is approximately coordinate-wise monotone in a suitable feature space, dependence-only alignment can remain stable even as marginals change. The protocol makes this contract auditable through diagnostic proxies, label-free calibration, and a fallback rule for batches outside the certified regime. Existing robustness efforts are often effective in specific settings. They usually do not provide a single operational contract that links invariance assumptions, measurable diagnostics, and deployment actions. SSCA connects multi-view alignment (Hotelling, 1936; Carroll, 1968; Andrew et al., 2013), transport-based alignment (Rabin et al., 2011; Cuturi, 2013; Peyré & Cuturi, 2019), and deployment-shift monitoring/calibration (Gama et al., 2014; Platt, 1999; Guo et al., 2017; Sun et al., 2020; Wang et al., 2021) into one auditable protocol. It also builds on canonical correlation methods (Hardoon et al., 2004; Bach & Jordan, 2002), optimal transport theory including Gromov-Wasserstein distances (Cuturi & Doucet, 2014; Mémoli, 2011; Peyré et al., 2016), robustness under distribution shift (Ovadia et al., 2019; Koh et al., 2021), and contrastive vision-language pretraining (Radford et al., 2021).

**Stable Spectral Copula Alignment (SSCA)** is a deployment protocol for estimating a copula-invariant dependence operator under a declared stability contract. The design principle is direct: to approximate the ideal operator $M_*$, the estimator must control the main error sources introduced by finite data, imperfect coupling, and numerical approximation. The perturbation $E = \widehat{M} - M_*$ is decomposed

[1]Tsinghua Shenzhen International Graduate School, Tsinghua University, Shenzhen, P.R. China. Correspondence to: Shao-Lun Huang <shaolun.huang@sz.tsinghua.edu.cn>.

*Proceedings of the 43rd International Conference on Machine Learning*, Seoul, South Korea. PMLR 306, 2026. Copyright 2026 by the author(s).

as

$$E = \underbrace{E_{\text{rank}}}_{\text{rank error}} + \underbrace{E_{\text{cpl}}}_{\text{coupling error}} + \underbrace{E_{\text{samp}} + E_{\text{num}}}_{\text{sampling \& numerical error}} . \quad (1)$$

Each module controls one part of this decomposition: (i) a **tie-aware, clipped rank-probit copula stabilizer** controls $\varepsilon_{\text{rank}}$ and provides finite-sample monotone-invariance bounds with tie and approximation terms; (ii) a **dependence-weighted sliced-Wasserstein hub coupling** controls $\varepsilon_{\text{cpl}}$, induces globally consistent multiway alignment, and enables cycle auditing; and (iii) a **diagonal-stabilized block-spectral learner** controls $\varepsilon_{\text{samp}}$ and $\varepsilon_{\text{num}}$ through eigengap-normalized diagnostics, optimizing for an identifiable shared subspace while monitoring numerical stability. These modules produce a unified **diagnostic dictionary** $\{\widehat{\varepsilon}_{\text{rank}}, \widehat{\varepsilon}_{\text{cpl}}, \widehat{\varepsilon}_{\text{samp}}, \widehat{\varepsilon}_{\text{num}}\}$, where each $\widehat{\varepsilon}_{\bullet}$ is an empirical proxy for the corresponding error $\varepsilon_{\bullet}$.

The core theoretical component is a **calibrated, actionable Davis–Kahan inequality** (Theorem 4.1) that bounds the subspace deviation $\| \sin \Theta(\widehat{U}, U_*) \|_F$ by a linear combination of observable proxies scaled by the empirical eigengap $\widehat{\gamma}$. The coefficients $c_{\text{rank}}, c_{\text{cpl}}, c_{\text{samp}}, c_{\text{num}}$ are not universal constants; they are deployment-local scaling factors calibrated from healthy unlabeled batches by quantile regression. The resulting inequality can be evaluated without target labels or access to the population subspace. It enables a **label-free monitoring gate**, $\text{Gate}(\tau, \gamma_{\min})$, which defines two deployment modes. **Stability Mode** ($\text{Gate} = 1$) permits alignment updates when the calibrated inequality indicates bounded subspace perturbation under in-scope, approximately monotone drift. **Fallback Mode** ($\text{Gate} = 0$) treats the batch as outside the certified regime, logs the dominant diagnostics, and reverts to a conservative no-update control instead of forcing an unsupported alignment update. The same diagnostics also drive a **budget-constrained remediation loop** that selects hyperparameter adjustments by minimizing diagnostic-weighted error under a computational budget. The empirical evidence is reported under two scopes. Controlled affine scaling tests the formal monotone contract. Raw-pipeline drifts, non-monotone folding/saturation, missing-modality protocols, and multi-backbone studies evaluate whether the same diagnostics provide useful behavior beyond the theorem assumptions. In these cases, SSCA provides a diagnostic safe-failure policy: the gate suppresses unsupported updates and uses a no-update fallback when the dependence structure is no longer identifiable. The main contributions are as follows:

**Auditable copula-stable contract.** Approximately coordinate-wise, orientation-preserving monotone marginal distortions are formalized, and clipped soft-rank Gaussianization is analyzed through finite-sample order-stability bounds with tie and approximation terms.

**Multiway coherence and measurable audits.** Dependence-weighted sliced-$W_2$ hub coupling promotes coherent multiway alignment and supports coupling/coherence audits; diagonal-stabilized block-spectral learning yields an eigengap-normalized Davis–Kahan subspace-risk inequality with measurable proxy terms.

**Calibrated gate, coverage, and trade-offs.** Label-free calibration sets the thresholds. A diagnostic gate triggers budgeted remediation or conservative no-update fallback when risk is detected. Coverage and trade-offs are quantified in Table 6.

**Evidence under controlled and deployment drifts.** SSCA reduces degradation under monotone scaling on CMU-MOSEI/MELD and improves frozen CLIP retrieval robustness on CC3M-500K. Raw-pipeline, multi-backbone, and compute-matched controls are summarized in Appendix Tables 21, 22, 23. Appendix Figure 4 gives an out-of-scope non-monotone case where forced Stability Mode yields large subspace error, while gate+fallback keeps degradation close to the no-update baseline.

## 2. Preliminaries and Problem Formulation

### 2.1. Multimodal Alignment Setting

Consider a setting with $d \geq 2$ modalities (e.g., text, image, audio). Let $\mathcal{X}^{(i)}$ denote the raw input space for modality $i$. A pre-trained encoder $f_i : \mathcal{X}^{(i)} \to \mathbb{R}^{p_i}$ maps inputs to feature vectors. Given a batch of $m$ paired samples, feature matrices are obtained as $H^{(i)} \in \mathbb{R}^{m \times p_i}$. The goal of multimodal alignment is to learn a set of projection matrices $W_i \in \mathbb{R}^{p_i \times k}$ (with $W_i^\top W_i = I_k$) that map features from each modality into a shared $k$-dimensional subspace where they are maximally correlated or similar. SSCA first applies a per-modality stabilizer $\mathcal{T}^{(i)}$ to the feature matrix and then projects the stabilized features, giving $Z_0^{(i)} = \mathcal{T}^{(i)}(H^{(i)})W_i$. In a deployment-realistic frozen-feature protocol, the encoders $f_i$ are fixed, and only the projections $W_i$ and alignment parameters are trainable.

### 2.2. Deployment Shift Model

Distribution shift encountered during deployment is modeled as per-coordinate marginal distortions that may affect each modality independently and potentially evolve over time. Formally, for modality $i$ at deployment time, observed features become $\widetilde{H}^{(i)} = \phi^{(i)}(H^{(i)})$, where $\phi^{(i)} : \mathbb{R}^{p_i} \to \mathbb{R}^{p_i}$ is a transformation applied coordinate-wise. The core operational scope of SSCA is defined for shifts where each $\phi^{(i)}$ is approximately coordinate-wise and orientation-preserving monotone. This class includes positive affine scaling, $x \mapsto ax + b$ with $a > 0$, and pointwise monotone nonlinearities such as $x \mapsto \log(1 + |x|)\text{sign}(x)$ and $x \mapsto \tanh(\beta x)$.

Approximate pipeline effects, including moderate compression, sensor gain changes, and certain normalization drifts, can behave as marginal feature shifts and are evaluated empirically. They are outside the exact invariance claim.

Shifts that are non-monotone (e.g., adversarial perturbations, certain stylizations), cause heavy ties (e.g., extreme quantization), or alter the semantic/content relationship are considered out-of-scope for the Stability Mode; SSCA's Fallback Mode defines conservative behavior for such cases.

### 2.3. Copula Invariance and Rank Maps

For a continuous random vector $(X^{(1)}, \ldots, X^{(d)})$ with joint CDF $F$ and marginal CDFs $F_i$, Sklar's Theorem states:

$$F(x^{(1)}, \ldots, x^{(d)}) = C(F_1(x^{(1)}), \ldots, F_d(x^{(d)})), \quad (2)$$

where $C : [0,1]^d \to [0,1]$ is the copula. A key property is that if $g_i : \mathbb{R} \to \mathbb{R}$ are strictly increasing, the copula of $(g_1(X^{(1)}), \ldots, g_d(X^{(d)}))$ is identical to that of $(X^{(1)}, \ldots, X^{(d)})$. Thus, dependence captured by $C$ is invariant to orientation-preserving monotone marginal transformations.

In practice, with finite samples, $F_i$ is approximated using ranks. For a vector $v \in \mathbb{R}^m$, the normalized empirical rank with stable tie-breaking is:

$$\mathrm{uRank}(v)_t = \frac{\mathrm{rank}(v_t) - \frac{1}{2}}{m} \in (0,1), \quad t = 1, \ldots, m, \quad (3)$$

where subtracting $1/2$ avoids boundary issues. For differentiable training, a smooth surrogate, the soft rank, is employed:

$$\mathrm{sRank}_\tau(v)_t = \frac{1}{m} \sum_{r=1}^{m} \sigma\left(\frac{v_t - v_r}{\tau}\right), \quad (4)$$

where $\sigma$ is the logistic sigmoid and $\tau > 0$ is a temperature parameter.

### 2.4. Sliced Wasserstein Distance and Barycenters

The sliced Wasserstein distance (Rabin et al., 2011) provides a computationally efficient approximation to the Wasserstein distance by projecting distributions onto random one-dimensional directions. For distributions $P, Q$ on $\mathbb{R}^k$:

$$\mathrm{SW}_2^2(P, Q) = \int_{\mathbb{S}^{k-1}} \mathcal{W}_2^2(P_\theta, Q_\theta) d\theta, \quad (5)$$

where $P_\theta$ is the pushforward of $P$ under projection onto $\theta \in \mathbb{S}^{k-1}$. In one dimension, $\mathcal{W}_2^2$ has a closed form using quantile functions: $\mathcal{W}_2^2(P_\theta, Q_\theta) = \int_0^1 (Q_{P_\theta}(u) - Q_{Q_\theta}(u))^2 du$. Given $d$ 1D distributions with quantile functions $Q_i$ and weights $w_i \geq 0$, $\sum_i w_i = 1$, the quantile function of their Wasserstein barycenter is $Q_B(u) = \sum_{i=1}^{d} w_i Q_i(u)$. This linearity supports the proposed multiway hub construction.

### 2.5. Spectral Perturbation Theory

Let $M_* \in \mathbb{R}^{n \times n}$ be a symmetric matrix with top-$k$ eigenspace spanned by the columns of $U_* \in \mathbb{R}^{n \times k}$ ($U_*^\top U_* = I_k$). Let $\widehat{M} = M_* + E$ be an empirical estimate. The Davis–Kahan theorem (Davis & Kahan, 1970; Stewart & Sun, 1990) bounds the distance between the eigenspaces:

$$\|\sin\Theta(\widehat{U}, U_*)\|_F \leq \frac{C_{\mathrm{DK}} \|E\|_2}{\gamma_*}, \quad (6)$$

where $\widehat{U}$ spans the top-$k$ eigenspace of $\widehat{M}$, $\gamma_* = \lambda_k(M_*) - \lambda_{k+1}(M_*) > 0$ is the eigengap, $\|\cdot\|_2$ is the spectral norm, and $C_{\mathrm{DK}}$ absorbs the norm convention and rank-dependent constants. SSCA makes this bound actionable by decomposing $\|E\|_2$ into measurable components.

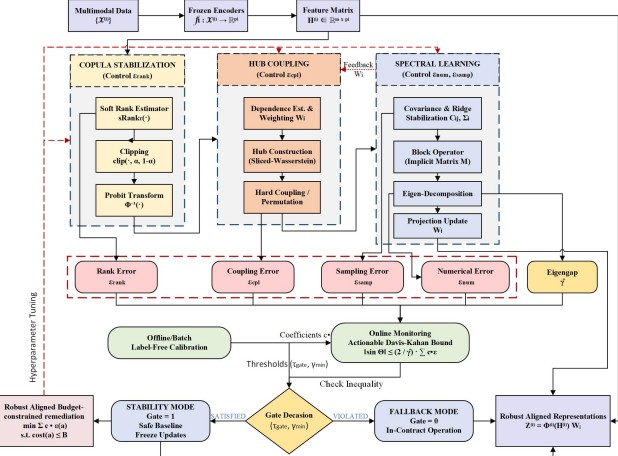

*Figure 1.* SSCA deployment protocol: three error-control modules control rank, coupling, sampling, and numerical errors, producing diagnostic proxies for the calibrated gate between Stability and Fallback Modes. The representation label in the monitoring path denotes the ideal shared subspace associated with $U_*$.

## 3. The SSCA Protocol: Error-Source Decomposition and Control

The SSCA protocol approximates an ideal copula-invariant dependence operator $M_*$ that provides stable alignment under in-scope monotone marginal shifts in the population setting. As illustrated in Figure 1, the protocol architecture controls the main error sources that cause the empirical operator $\widehat{M}$ to deviate from $M_*$, as formalized in Eq. (1). This decomposition enables independent auditing and targeted remediation for each error type.

### 3.1. Module 1: Copula Stabilization (Control of $\varepsilon_{\mathrm{rank}}$)

The ideal Gaussianization map for a scalar feature $x$ with CDF $F$ is $\Phi^{-1}(F(x))$, where $\Phi$ is the standard normal CDF. This yields a representation invariant to orientation-

preserving monotone transforms of $x$. A finite-sample, differentiable approximation with explicit error control is implemented.

**Differentiable Rank-Probit with Clipping.** For a batch of features for modality $i$, $H^{(i)} \in \mathbb{R}^{m \times p_i}$, per-coordinate stabilization is applied:

$$G_{:,j}^{(i)} = \Phi^{-1} \left( \text{clip} \left( \text{sRank}_{\tau_{\text{rank}}}(H_{:,j}^{(i)}), \alpha, 1 - \alpha \right) \right), \quad (7)$$

Here, $j = 1, \ldots, p_i$, $\text{sRank}_{\tau_{\text{rank}}}$ is the soft rank for differentiability, $\text{clip}(u, \alpha, 1-\alpha)$ restricts input to $[\alpha, 1-\alpha]$, and $\Phi^{-1}$ is the probit function. Clipping is essential to bound the Lipschitz constant of $\Phi^{-1}$ on the interval, preventing explosion from extreme quantiles:

$$L_{\Phi^{-1}}(\alpha) = \frac{1}{\varphi(\Phi^{-1}(\alpha))} < \infty, \quad (8)$$

where $\varphi$ is the standard normal density. The value $\alpha = 0.005$ is used, giving $L_{\Phi^{-1}} \approx 69.0$.

**Finite-Sample Monotone Invariance Bound.**

**Theorem 3.1** (Monotone Invariance with Approximation Error). *Let $v \in \mathbb{R}^m$ be a batch of scalar features with empirical CDF $F$. Let $\widetilde{F}$ be an approximation to $F$ (e.g., based on $\text{sRank}_\tau$). Define the stabilized transform $\phi(x) = \Phi^{-1}(\text{clip}(\widetilde{F}(x), \alpha, 1 - \alpha))$. For any strictly increasing function $g$,*

$$|\phi(g(x)) - \phi(x)| \leq L_{\Phi^{-1}}(\alpha) \left( 2 \sup_{z \in \mathbb{R}} |\widetilde{F}(z) - F(z)| + \delta_{\text{tie}}(v, g) \right)$$
(9)

*where $\delta_{\text{tie}}(v, g) \leq \max_u \frac{\#\{t : v_t = u\}}{m} + \max_u \frac{\#\{t : g(v_t) = u\}}{m}$ captures the additional CDF mismatch due to ties in $v$ or $g(v)$.*

The proof (see Appendix F.3) follows from the Lipschitz property, the triangle inequality, and analyzing CDF changes under orientation-preserving monotone transforms. This theorem formalizes the contract: if rank approximation error and tie rate are small, the stabilization is approximately invariant.

**Rank Error Proxy and Calibration.** The rank approximation error is monitored via the proxy:

$$\widehat{\varepsilon}_{\text{rank}} = \max_{i,j} \frac{1}{m} \|\text{sRank}_{\tau_{\text{rank}}}(H_{:,j}^{(i)}) - \text{uRank}(H_{:,j}^{(i)})\|_1. \quad (10)$$

During training, a lightweight regularization term $\mathcal{L}_{\text{rank}} = \lambda_{\text{rank}} \cdot \widehat{\varepsilon}_{\text{rank}}$ is added to keep the soft ranks calibrated to the true ranks, ensuring the invariance bound remains tight.

## 3.2. Module 2: Dependence-Weighted Hub Coupling (Control of $\varepsilon_{\text{cpl}}$)

Aligning modalities in a pairwise manner leads to cycle inconsistencies. SSCA enforces global coherence by aligning all modalities to a common *hub* ordering derived from a sliced-Wasserstein barycenter.

**Preliminary Projections and Dependence Estimation.** Using current projection matrices $W_i$, preliminary aligned features are computed: $Z_0^{(i)} = G^{(i)} W_i \in \mathbb{R}^{m \times k}$. To weigh modalities according to their relational signal, pairwise dependence is estimated using Kendall's rank correlation $\tau$, which tracks monotone relations up to orientation. For efficiency, estimation is performed via random projections:

$$\widehat{\tau}_{ij} = \frac{1}{S_\tau} \sum_{s=1}^{S_\tau} \tau \left( Z_0^{(i)} \theta_s^{(\tau)}, Z_0^{(j)} \theta_s^{(\tau)} \right), \quad \theta_s^{(\tau)} \sim \text{Uniform}(\mathbb{S}^{k-1}),$$
(11)

where $\tau(u, v)$ computes the correlation between vectors $u, v \in \mathbb{R}^m$.

**Weight Construction.** From these estimates, symmetric pairwise weights that emphasize strong positive dependence are constructed:

$$\omega_{ij} = \frac{\exp([\widehat{\tau}_{ij}]_+/\tau_{\text{dep}})}{\sum_{1 \leq a < b \leq d} \exp([\widehat{\tau}_{ab}]_+/\tau_{\text{dep}})}, \quad (12)$$

where $\omega_{ji} = \omega_{ij}$, $\omega_{ii} = 0$, $[x]_+ = \max(x, 0)$. Modality importance weights for the barycenter are:

$$w_i = \frac{\sum_{j \neq i} [\widehat{\tau}_{ij}]_+}{\sum_{\ell=1}^d \sum_{j \neq \ell} [\widehat{\tau}_{\ell j}]_+}, \quad \sum_{i=1}^d w_i = 1. \quad (13)$$

**Hub Construction and Hard Coupling.** For $S$ random directions $\theta_s \in \mathbb{S}^{k-1}$, 1D projections $p_s^{(i)} = Z_0^{(i)} \theta_s$ are computed. Let $Q_s^{(i)}$ be the empirical quantile function of $p_s^{(i)}$. The hub quantile function for direction $s$ is:

$$Q_s^{(B)}(u) = \sum_{i=1}^d w_i Q_s^{(i)}(u), \quad u \in (0, 1). \quad (14)$$

For each modality, the discrepancy to the hub barycenter is computed from sorted projections:

$$\beta_i = \frac{1}{S} \sum_{s=1}^S \frac{1}{m} \sum_{t=1}^m \left( p_{s,(t)}^{(i)} - Q_s^{(B)}(u_t) \right)^2, \quad u_t = \frac{t - \frac{1}{2}}{m}. \quad (15)$$

The hub modality is chosen as $i^\star = \arg\min_i \beta_i$. For each direction, one-dimensional OT matches sorted samples, yielding a permutation matrix $\Pi_s^{(i \to i^\star)}$ between modality $i$ and the hub. The averaged directional plan $\widetilde{P}_i = S^{-1} \sum_s \Pi_s^{(i \to i^\star)}$ is rounded to a hard permutation $\Pi_i$ by assignment, with $\Pi_{i^\star} = I_m$. Aligned features are then $\widetilde{G}^{(i)} = \Pi_i G^{(i)}$ (row reordering). Soft coupling uses Sinkhorn normalization in ablations and is reported separately in the appendix.

**Coupling Error Proxies ($\widehat{\varepsilon}_{\mathrm{cpl}}$).** Proxies to quantify deviations from a stable coupling are introduced:

$$\widehat{\varepsilon}_{\mathrm{rnd}} = \max_i \frac{\|\Pi_i - \widetilde{P}_i\|_F}{\sqrt{m}}, \tag{16}$$

$$\widehat{\varepsilon}_{\mathrm{hub}} = \frac{\beta_{(2)} - \beta_{(1)}}{\beta_{(1)} + \epsilon_{\mathrm{hub}}}, \tag{17}$$

$$\widehat{\varepsilon}_{\mathrm{cpl}} = \widehat{\varepsilon}_{\mathrm{rnd}} + \frac{1}{\widehat{\varepsilon}_{\mathrm{hub}} + \epsilon_{\mathrm{hub}}} + c_{\mathrm{tie}}\widehat{\rho}_{\mathrm{tie}}. \tag{18}$$

Here, $\Pi_i$ is the hard coupling matrix, $\widetilde{P}_i$ is the averaged directional plan, $\beta_{(1)}$ and $\beta_{(2)}$ are the smallest and second-smallest hub discrepancies, $\epsilon_{\mathrm{hub}} > 0$ is a fixed numerical constant, and $\widehat{\rho}_{\mathrm{tie}}$ is the empirical tie rate from the stabilization step. These terms capture directional disagreement, hub ambiguity, and ordering ambiguity caused by ties.

### 3.3. Module 3: Diagonal-Stabilized Spectral Learning (Control of $\varepsilon_{\mathrm{samp}}$ and $\varepsilon_{\mathrm{num}}$)

Given the aligned features $\widetilde{G}^{(i)}$, this module learns the projection matrices $W_i$ by identifying the dominant shared subspace via a generalized eigenvalue problem, stabilized for finite-sample and numerical errors.

**Covariance Computation and Ridge Stabilization.** Features are centered: $\bar{G}^{(i)} = \widetilde{G}^{(i)} - \frac{1}{m}\mathbf{1}\mathbf{1}^\top \widetilde{G}^{(i)}$. Cross-covariances and ridge-regularized within-covariances are computed:

$$C_{ij} = \tfrac{1}{m}\bar{G}^{(i)\top}\bar{G}^{(j)} \ (i \neq j), \quad \Sigma_i = \tfrac{1}{m}\bar{G}^{(i)\top}\bar{G}^{(i)} + \lambda_{\mathrm{stab}}I_{p_i}. \tag{19}$$

The ridge term $\lambda_{\mathrm{stab}} > 0$ ensures invertibility and controls condition number. Whitening matrices are defined as $R_i = \Sigma_i^{-1/2}$.

**Block Operator and Spectral Objective.** A symmetric block matrix $\mathcal{M} \in \mathbb{R}^{p \times p}$ is constructed, where $p = \sum_{i=1}^d p_i$:

$$\mathcal{M}_{ij} = \begin{cases} \omega_{ij}R_i C_{ij}R_j, & i \neq j, \\ 0, & i = j. \end{cases} \tag{20}$$

The alignment objective is to maximize the sum of weighted correlations:

$$\max_{\{W_i^\top \Sigma_i W_i = I_k\}} \sum_{1 \leq i < j \leq d} \omega_{ij}\mathrm{tr}\left(W_i^\top C_{ij}W_j\right). \tag{21}$$

**Lemma 3.2** (Block-Spectral Relaxation). *Under the whitening reparameterization $W_i = R_i U_i$, the relaxed weighted trace objective in Eq. (21) is solved by the top-$k$ eigenspace of $\mathcal{M}$.*

*Proof.* (Sketch) The constraint $W_i^\top \Sigma_i W_i = I_k$ becomes $U_i^\top U_i = I_k$ under $W_i = R_i U_i$. Substituting into the objective yields $\sum_{i<j} \omega_{ij}\mathrm{tr}(U_i^\top R_i C_{ij}R_j U_j)$. Stacking $U = [U_1^\top \cdots U_d^\top]^\top$ rewrites the relaxed block trace as $\mathrm{tr}(U^\top \mathcal{M}U)$ up to the symmetric off-diagonal factor. By the Ky Fan theorem, the optimal relaxed subspace is spanned by the top-$k$ eigenvectors of $\mathcal{M}$. $\square$

**Efficient Eigensolver and Projection Update.** The top-$k$ eigenvectors $\widehat{U} = [\widehat{U}_1^\top \cdots \widehat{U}_d^\top]^\top$ of $\mathcal{M}$ are computed using an iterative method (e.g., LOBPCG). Matrix-vector products with $\mathcal{M}$ are computed implicitly without forming the $p \times p$ matrix explicitly, using the factored form:

$$(\mathcal{M}x)_i = \sum_{j \neq i} \omega_{ij}R_i \left(C_{ij}(R_j x_j)\right), \tag{22}$$

where $x = [x_1^\top \cdots x_d^\top]^\top$. This reduces complexity to $O(d^2 mk)$ after projection/cached minibatch matvecs. The projection matrices are updated as:

$$W_i \leftarrow \mathrm{qf}(R_i\widehat{U}_i), \tag{23}$$

where $\mathrm{qf}(\cdot)$ extracts an orthonormal basis via QR decomposition.

**Spectral Error Proxies.** Two key proxies are monitored:

$$\widehat{\varepsilon}_{\mathrm{samp}} = \max_{i \neq j} \|C_{ij} - C_{ij}^{(\mathrm{EMA})}\|_F, \tag{24}$$

$$\widehat{\varepsilon}_{\mathrm{num}} = \frac{\|\mathcal{M}\widehat{U} - \widehat{U}\widehat{\Lambda}\|_F}{\|\widehat{U}\|_F}, \tag{25}$$

where $C_{ij}^{(\mathrm{EMA})}$ is an exponential moving average of past cross-covariances, capturing distribution drift. $\widehat{\Lambda} = \widehat{U}^\top \mathcal{M}\widehat{U}$ is the Ritz value matrix. $\widehat{\varepsilon}_{\mathrm{samp}}$ tracks sampling instability, while $\widehat{\varepsilon}_{\mathrm{num}}$ quantifies the eigensolver's residual error. The empirical eigengap $\widehat{\gamma} = \lambda_k(\mathcal{M}) - \lambda_{k+1}(\mathcal{M})$ is a direct stability indicator.

## 4. Stability Contract and Actionable Diagnostics

### 4.1. Calibrated Actionable Davis–Kahan Inequality

The classical Davis–Kahan bound is non-operational because $\|E\|_2$ is unknown. A practical version is derived by linking $\|E\|_2$ to the proposed observable diagnostic proxies.

**Theorem 4.1** (Calibrated Actionable Inequality). *Let $M_*$ be the ideal population operator (under perfect stabilization and coupling) with top-$k$ eigenspace $U_*$ and eigengap $\gamma_* > 0$. Let $\widehat{M}$ be the empirical operator constructed by SSCA with diagnostics $\widehat{\varepsilon}_\bullet$ and eigengap $\widehat{\gamma}$. Under assumptions of bounded feature moments (see Appendix for formal*

*statements) and stable coupling weights, there exist* calibratable *non-negative coefficients* $c_{\text{rank}}, c_{\text{cpl}}, c_{\text{samp}}, c_{\text{num}}$ *such that, with high probability,*

$$\|\sin\Theta(\widehat{U}, U_*)\|_F \leq \frac{c_{\text{rank}}\widehat{\varepsilon}_{\text{rank}} + c_{\text{cpl}}\widehat{\varepsilon}_{\text{cpl}} + c_{\text{samp}}\widehat{\varepsilon}_{\text{samp}} + c_{\text{num}}\widehat{\varepsilon}_{\text{num}}}{\widehat{\gamma} + 10^{-8}} \quad (26)$$

*The coefficients absorb intrinsic problem constants, including the Lipschitz factor* $L_{\Phi^{-1}}(\alpha)$*, spectral bounds in the factor model, coupling stability constants, and the Davis–Kahan norm convention. They can be estimated from unlabeled data without knowing* $U_*$*.*

*Proof.* (Sketch) The proof proceeds in three steps. First, the perturbation $E = \widehat{M} - M_*$ is expressed in terms of the algorithmic errors: $E = E_{\text{rank}} + E_{\text{cpl}} + E_{\text{samp}} + E_{\text{num}}$, corresponding to the modules. Second, the spectral norm of each component is bounded using the corresponding proxy. For example, $\|E_{\text{rank}}\|_2 \leq c_{\text{rank}} \cdot \widehat{\varepsilon}_{\text{rank}}$, where $c_{\text{rank}}$ involves the Lipschitz constant $L_{\Phi^{-1}}(\alpha)$ and the operator norm of the weight matrices. Similar bounds hold for other components, with coefficients involving the modality weights $\omega_{ij}$ and spectral bounds of the covariance matrices. Third, the triangle inequality and the Davis–Kahan theorem are applied. The final coefficients are the aggregates from these bounds. The calibration protocol provides a data-driven way to estimate the relative scales of these aggregate coefficients.

## 4.2. Label-Free Calibration Protocol

The coefficients in Theorem 4.1 can be estimated using a small set of "healthy" batches (e.g., from a validation set or a stable period of deployment) without any labeled performance degradation data.

**Collect Calibration Data:** For $N$ healthy batches, compute the diagnostic vector $\widehat{\varepsilon}^{(n)} = [\widehat{\varepsilon}_{\text{rank}}^{(n)}, \widehat{\varepsilon}_{\text{cpl}}^{(n)}, \widehat{\varepsilon}_{\text{samp}}^{(n)}, \widehat{\varepsilon}_{\text{num}}^{(n)}]$ and eigengap $\widehat{\gamma}^{(n)}$ for each batch $n = 1, \ldots, N$.

**Generate Reference Distances:** For each batch, compute a reference subspace distance $d^{(n)}$. Since $U_*$ is unknown, a surrogate is used: the distance between subspaces estimated from two independent halves of a large held-out batch, or the distance between consecutive batches in a stable stream. This surrogate estimates subspace variation under healthy operation.

**Solve Quantile Regression:** Estimate coefficient ratios by solving:

$$\min_{c \geq 0} \sum_{n=1}^{N} \rho_\tau \left( \widehat{\gamma}^{(n)} d^{(n)} - \sum_{\bullet} c_{\bullet} \widehat{\varepsilon}_{\bullet}^{(n)} \right), \quad (27)$$

where $\rho_\tau(u) = u(\tau - \mathbb{I}\{u < 0\})$ is the check function for quantile $\tau$ (e.g., $\tau = 0.9$). This fits the coefficients so that

the weighted proxy sum forms an upper envelope for the scaled reference distance.

**Set Monitoring Thresholds:** Compute the calibrated risk values $\widehat{\Delta}^{(n)} = (\sum_{\bullet} c_{\bullet} \widehat{\varepsilon}_{\bullet}^{(n)})/(\widehat{\gamma}^{(n)} + 10^{-8})$ on the healthy batches. Set $\tau_{\text{gate}}$ to a high quantile (e.g., 95th) of these values, and set $\gamma_{\min}$ as the $\delta$-quantile (e.g., 5th) of $\widehat{\gamma}^{(n)}$.

These coefficients are deployment-local scaling factors rather than universal constants. A change in backbone, feature normalization, modality set, or batching policy can alter the relation between proxy magnitudes and subspace deviation. The intended use is therefore to recalibrate coefficients and thresholds on a short healthy unlabeled stream after such changes, while keeping the SSCA objective fixed. This preserves auditability: each rejection is traced to one of the four proxy families and the eigengap margin, not an opaque confidence score.

## 4.3. Stability Gate and Two-Part Contract

The calibrated inequality enables a data-driven monitoring gate:

$$\text{Gate}(\tau_{\text{gate}}, \gamma_{\min}) = \mathbb{I}\left[ \frac{\sum_{\bullet} c_{\bullet} \widehat{\varepsilon}_{\bullet}}{\widehat{\gamma} + 10^{-8}} \leq \tau_{\text{gate}} \ \wedge \ \widehat{\gamma} \geq \gamma_{\min} \right]. \quad (28)$$

This gate triggers the corresponding contract mode:

**Stability Mode (Gate = 1):** The batch lies within the calibrated stability envelope. For in-scope orientation-preserving monotone drift, the actionable inequality provides a checkable upper envelope on the subspace error at the calibrated tolerance level. The standard SSCA update is allowed.

**Fallback Mode (Gate = 0):** The batch is treated as outside the certified regime. SSCA suppresses further alignment updates under the unsupported contract, logs the dominant diagnostics, and uses the corresponding compute-matched no-update control. This is a policy-level safe-failure rule and does not guarantee accuracy outside the declared assumptions. A deployment may additionally issue an alert or freeze optional adaptation when instability is severe.

A detailed scope table (provided in Appendix K) lists in-scope conditions (monotone drift, low tie rate, sufficient batch statistics) and out-of-scope conditions (non-monotone drift, heavy ties, weak dependence, small batch), along with the expected diagnostic signature and triggered action.

## 4.4. Budget-Constrained Remediation Loop

When diagnostics approach the calibrated threshold, or when a monitored task metric declines in settings where labels are available, SSCA initiates bounded remediation before full Fallback Mode activation. Remediation is formulated as a budget-constrained optimization problem. Let

$\mathcal{A}$ denote feasible actions: increasing the soft-rank temperature $\tau_{\text{rank}}$, expanding the number of slicing directions $S$, adjusting the ridge parameter $\lambda_{\text{stab}}$, or enlarging sketch size for streaming stabilization. Each action $a \in \mathcal{A}$ incurs computational cost $\text{cost}(a)$ (e.g., FLOPs, memory). Applying $a$ yields updated diagnostics $\widehat{\varepsilon}_{\bullet}(a)$. The remediation objective is:

$$\min_{a \in \mathcal{A}} \sum_{\bullet} c_{\bullet} \widehat{\varepsilon}_{\bullet}(a) \quad \text{subject to} \quad \text{cost}(a) \leq B, \quad (29)$$

where $B$ is a predefined compute budget. This formulation ensures that remediation is both *diagnostic-driven* (targeting the dominant error source) and *resource-efficient*. In practice, the solution is approximated by evaluating a small set of candidate actions. If the recalculated gate remains inactive, the batch uses Fallback Mode.

# 5. Experimental Evaluation

## 5.1. Experimental Setup

**Datasets and Tasks.** Four benchmarks are used: **CMU-MOSEI** (Bagher Zadeh et al., 2018) and **MELD** (Poria et al., 2019) for trimodal sentiment/emotion classification, **MSCOCO** (Lin et al., 2014) for image–text retrieval, and **CC3M-500K** (Sharma et al., 2018) for large-scale image–text retrieval. For retrieval tasks, frozen CLIP ViT-B/16 features are used unless otherwise stated.

**Protocol and Perturbations.** A frozen-feature protocol is adopted: pretrained encoders are fixed, and only projection heads and alignment parameters are trained. Robustness is evaluated under two categories of perturbations. The first category uses controlled feature-space distortions, including the formal monotone case of per-dimension affine scaling with magnitude $10\times$, together with JPEG compression as an approximate pipeline stress test. The second category uses deployment-style drifts, including JPEG compressor changes, tokenizer changes, and audio codec changes. Raw-pipeline drifts are treated as empirical stress tests, not as exact invariance cases.

**Baselines.** The baselines cover contrastive learning methods (InfoNCE (Chen et al., 2020), SigLIP (Zhai et al., 2023)), correlation-based methods (DCCA (Andrew et al., 2013), GCCA, CCA), optimal-transport alignment (SW-Align, Sinkhorn), and recent robust multimodal models (MMML (Wu et al., 2024), MUG (Mai et al., 2024), EMT (Sun et al., 2024), CRNet (Shi et al., 2024)). Two targeted baseline families are included: (i) **Copula-only + Strong Objective**, which applies the copula rank stabilizer followed by InfoNCE or SigLIP, and (ii) **Pairwise OT + Cycle Correction**, which uses pairwise sliced-Wasserstein alignment with a cycle-consistency loss $\mathcal{L}_{\text{cycle}}$. All baselines use the same frozen features, validation protocol, and projection-level compute budget; Appendix C.5 provides the closed-

loop baseline specification.

**Metrics.** For a clean score $s_{\text{clean}}$ and perturbed score $s_{\pi}$,

$$\Delta(\pi) = s_{\text{clean}} - s_{\pi}, \quad (30)$$

$$\text{Red}(\pi) = 100 \cdot \frac{\Delta_{\text{base}}(\pi) - \Delta_{\text{SSCA}}(\pi)}{\Delta_{\text{base}}(\pi)}. \quad (31)$$

Both absolute scores and degradation metrics are reported. Calibration quantities are reported where applicable. For streaming drift, AUROC, lead time, and recovery are reported.

**Implementation Details.** Unless specified otherwise, $k = 128$, $m = 256$, $\tau_{\text{rank}} = 0.1$, $\alpha = 0.005$, and $\lambda_{\text{stab}} = 0.01$ are used with AdamW learning rate $10^{-4}$. Results are averaged over 5 seeds, except for the CC3M-500K CLIP sweep, which uses 3 seeds. Full protocol and hyperparameter details are provided in Appendices I and J.10.

## 5.2. Main Results: Robustness under Monotone Distortions

Table 1 reports CMU-MOSEI binary accuracy and MELD overall accuracy under the severe $10\times$ affine-scaling perturbation. SSCA achieves the highest clean accuracy and the smallest degradation $\Delta$ on both datasets. MMML is the strongest general baseline, but SSCA reduces its degradation by about 44% on both CMU-MOSEI and MELD. Copula-only + SigLIP improves substantially over vanilla SigLIP, showing that copula stabilization removes much of the marginal sensitivity. Its degradation remains 0.5–0.7 points higher than full SSCA, indicating that hub coupling and spectral stabilization provide additional robustness beyond marginal stabilization.

*Table 1.* Performance under $10\times$ affine scaling. Degradation $\Delta = \text{Clean} - \text{Scaled}$. †: $p < 0.05$ vs. best baseline (MMML).

| Method | CMU-MOSEI (Acc-2 %) | | | MELD (Acc %) | | |
|---|---|---|---|---|---|---|
| | Clean | Scaled | $\Delta \downarrow$ | Clean | Scaled | $\Delta \downarrow$ |
| InfoNCE | 78.9±0.5 | 72.4±0.7 | 6.5±0.4 | 59.8±0.6 | 55.0±0.7 | 4.8±0.3 |
| SigLIP | 79.5±0.5 | 73.2±0.7 | 6.3±0.4 | 60.3±0.6 | 55.6±0.7 | 4.7±0.3 |
| DCCA | 76.8±0.6 | 73.5±0.6 | 3.3±0.2 | 58.5±0.7 | 56.1±0.6 | 2.4±0.2 |
| GCCA | 77.6±0.6 | 74.7±0.6 | 2.9±0.2 | 59.0±0.7 | 56.7±0.6 | 2.3±0.2 |
| SW-Align | 79.2±0.5 | 75.9±0.6 | 3.3±0.2 | 60.7±0.6 | 58.0±0.6 | 2.7±0.2 |
| Sinkhorn | 78.8±0.5 | 75.3±0.6 | 3.5±0.2 | 60.1±0.6 | 57.4±0.6 | 2.7±0.2 |
| MUG | 87.6±0.3 | 83.8±0.4 | 3.8±0.2 | 66.2±0.5 | 63.5±0.5 | 2.7±0.2 |
| EMT | 86.0±0.4 | 82.5±0.5 | 3.5±0.3 | 62.5±0.5 | 59.8±0.6 | 2.7±0.2 |
| CRNet | 86.1±0.4 | 83.0±0.5 | 3.1±0.2 | 63.2±0.5 | 60.5±0.5 | 2.7±0.2 |
| MMML | 86.7±0.3 | 83.5±0.4 | 3.2±0.2 | 66.7±0.4 | 64.0±0.4 | 2.7±0.1 |
| **Copula-only + SigLIP** | 87.2±0.3 | 84.9±0.3 | 2.3±0.1 | 67.0±0.4 | 64.8±0.4 | 2.2±0.1 |
| **SSCA** | **87.8±0.3†** | **86.0±0.3†** | **1.8±0.1** | **67.8±0.4†** | **66.3±0.4†** | **1.5±0.1** |

## 5.3. Critical Baseline Analysis

Critical baselines test whether copula stabilization or pairwise OT with cycle correction can explain the robustness gains. Under $10\times$ scaling, Copula-only + SigLIP reduces degradation over plain contrastive learning, while SSCA further improves the degradation on both datasets (CMU-MOSEI: $\Delta$ 1.8 vs. 2.3; MELD: $\Delta$ 1.5 vs. 2.2). Cop-

ula stabilization removes marginal scale sensitivity, but it does not enforce globally coherent multiway coupling or protect the shared subspace against sampling and numerical instability. The hub term aligns modalities through a dependence-weighted barycentric reference. The spectral term uses an eigengap-aware objective and exposes residual sampling/numerical error to the gate. Full numbers are reported in Appendix Table 20.

The comparison separates three roles: rank stabilization removes marginal scale sensitivity, hub coupling enforces cross-modal consistency beyond independent pairwise matches, and spectral stabilization makes the shared subspace identifiable enough for the gate to monitor residual instability.

### 5.4. Large-Scale Retrieval with Frozen CLIP

Table 2 reports CC3M-500K image–text retrieval with a frozen CLIP ViT-B/16 backbone under JPEG compression. SSCA improves clean Recall@1 and reduces degradation as compression increases. At Q=50, the reported degradation reduction is 52.9%, computed from the unrounded degradation estimates. These results indicate that the diagnostic protocol remains useful when SSCA is attached to a fixed, pretrained backbone on large-scale web-curated data.

*Table 2.* Image-text retrieval on CC3M-500K with frozen CLIP ViT-B/16 (Recall@1). $\text{Red}(Q)$ denotes the percentage degradation reduction at quality $Q$ compared to the CLIP Baseline. $\Delta$ and Red are computed before rounding displayed Recall@1 values. †: $p < 0.05$ vs. CLIP+MMML.

| Method | Clean | Q=70 | Q=50 | Q=30 | $\Delta$ @Q=50 | Red(50) |
|---|---|---|---|---|---|---|
| CLIP Baseline | 58.7±0.5 | 56.2±0.6 | 53.6±0.6 | 49.1±0.7 | 5.1±0.3 | – |
| CLIP + SigLIP | 59.3±0.5 | 57.0±0.5 | 54.5±0.6 | 50.2±0.7 | 4.8±0.3 | 5.9% |
| CLIP + SW-Align | 58.9±0.5 | 56.8±0.6 | 54.2±0.6 | 50.0±0.7 | 4.7±0.3 | 7.8% |
| CLIP + MUG | 59.5±0.5 | 57.3±0.5 | 55.0±0.6 | 51.0±0.6 | 4.5±0.3 | 11.8% |
| CLIP + MMML | 59.8±0.4 | 57.5±0.5 | 55.9±0.5 | 52.1±0.6 | 3.9±0.2 | 23.5% |
| **SSCA (CLIP)** | **59.8**±0.4 | **58.4**±0.5 | **57.3**±0.5† | **54.9**±0.6† | **2.4**±0.2 | **52.9%** |

### 5.5. Real-World Deployment Drift and Generalization

Table 3 reports deployment-style drifts on CMU-MOSEI and MELD. SSCA achieves the lowest degradation across all drift types. On MELD, the average $\Delta$ is 1.4, a 33% reduction relative to MMML (avg. $\Delta = 2.1$). On CMU-MOSEI, the average $\Delta$ is 1.7, about a 28% reduction relative to MMML (avg. $\Delta = 2.4$). Across all six drift cases, SSCA reports an average $\Delta$ of 1.6. These results are empirical stress tests beyond the formal monotone contract, not an extension of the theorem. Adapter-based fine-tuning results are reported in Appendix O.5.

**Additional Controls.** Additional raw-pipeline, multi-backbone/task, and compute-matched controls are reported in Appendix Tables 21, 22, and 23.

*Table 3.* Degradation ($\Delta$) under real-world deployment drifts on CMU-MOSEI and MELD. Lower is better.

| Method | CMU-MOSEI $\Delta \downarrow$ | | | MELD $\Delta \downarrow$ | | | Avg. $\Delta$ |
|---|---|---|---|---|---|---|---|
| | JPEG | Tokenizer | Audio | JPEG | Tokenizer | Audio | |
| InfoNCE | 3.5±0.3 | 3.1±0.3 | 3.8±0.3 | 3.2±0.3 | 2.8±0.2 | 3.5±0.3 | 3.3±0.2 |
| SigLIP | 3.3±0.3 | 2.9±0.3 | 3.6±0.3 | 3.0±0.3 | 2.6±0.2 | 3.3±0.3 | 3.1±0.2 |
| MUG | 2.4±0.2 | 2.1±0.2 | 2.6±0.2 | 2.3±0.2 | 2.0±0.2 | 2.5±0.2 | 2.3±0.1 |
| EMT | 2.7±0.2 | 2.4±0.2 | 3.0±0.2 | 2.5±0.2 | 2.2±0.2 | 2.8±0.2 | 2.6±0.1 |
| CRNet | 2.5±0.2 | 2.2±0.2 | 2.8±0.2 | 2.3±0.2 | 2.0±0.2 | 2.5±0.2 | 2.4±0.1 |
| MMML | 2.4±0.2 | 2.1±0.2 | 2.6±0.2 | 2.1±0.2 | 1.9±0.2 | 2.3±0.2 | 2.2±0.1 |
| **SSCA** | **1.7**±0.1 | **1.5**±0.1 | **1.9**±0.1 | **1.4**±0.1 | **1.2**±0.1 | **1.6**±0.1 | **1.6**±0.1 |

### 5.6. Diagnostic Calibration and Streaming Remediation

**Calibration Verification.** On synthetic data with known latent subspace perturbations, the proxy inequality in Eq. (26) is compared with the true principal-angle error. Appendix Figure 2 reports strong agreement between the proxy and true error (Pearson $r = 0.967$, Spearman $\rho = 0.941$) and 95.0% empirical coverage for the calibrated envelope.

**Streaming Drift Detection.** A gradual scaling drift is simulated on MELD and CMU-MOSEI, with scale factor increasing from 0.5 to 20 over 100 batches. Table 4 reports detection performance for individual diagnostics and their logistic combination on MELD. AUROC and lead time are computed post hoc against the drift indicator. The combined diagnostic achieves AUROC 0.94 and an average lead time of 7 batches before a significant performance drop. The label-free gate is defined in Appendix K.

*Table 4.* Streaming drift detection performance on MELD under gradual scaling drift.

| Diagnostic | AUROC ↑ | AUPRC ↑ | Avg. Lead (batches) | FPR@95% TPR ↓ |
|---|---|---|---|---|
| $\hat{\varepsilon}_{\text{rank}}$ | 0.88±0.03 | 0.72±0.04 | 6.3±0.8 | 0.12±0.02 |
| $\hat{\varepsilon}_{\text{cpl}}$ | 0.91±0.02 | 0.76±0.03 | 5.1±0.6 | 0.09±0.02 |
| $\hat{\varepsilon}_{\text{samp}}$ | 0.85±0.03 | 0.69±0.04 | 4.2±0.7 | 0.15±0.03 |
| Combined (Logistic) | **0.94**±0.02 | **0.81**±0.03 | **7.0**±0.7 | **0.06**±0.01 |

**Remediation Effectiveness.** Under abrupt $10\times$ scaling drift, a diagnostic-driven remediation strategy is evaluated on MELD and CMU-MOSEI. Table 5 reports the number of batches required to recover to 95% of pre-drift accuracy. SSCA diagnostic-driven remediation recovers within $11 \pm 2$ batches on MELD and $13 \pm 3$ batches on CMU-MOSEI. This is 45% and 43% faster than blind random adjustment on the two datasets, with lower overhead in the measured protocol. The results show that the diagnostics provide useful information for targeted recovery.

*Table 5.* Remediation under abrupt $10\times$ scaling drift. Batches required to reach 95% of pre-drift accuracy.

| Remediation Strategy | Batches to 95% Recovery ↓ | |
|---|---|---|
| | MELD | CMU-MOSEI |
| No remediation | 34 ± 4 | 38 ± 5 |
| Blind random adjustment | 20 ± 3 | 23 ± 4 |
| Scheduled adjustment (every 5 batches) | 18 ± 3 | 21 ± 3 |
| **SSCA Diagnostic-Driven** | **11 ± 2** | **13 ± 3** |

## 5.7. Gate Coverage and Trade-offs

The diagnostic gate is evaluated on end-to-end drifts. Table 6 reports how often the policy triggers conservative fallback (*Refuse*), how often an accepted update is empirically worse than fallback (*Unsafe Accept*), the opportunity cost of refusing when SSCA would help (*Regret*), and the net robustness gain (*Net Benefit*). Definitions, tolerance, and threshold details are in Appendix K.3. The gate keeps Stability Mode active for 81–88% of episodes, maintains a low unsafe-accept rate, and yields consistent net benefit.

*Table 6.* Gate coverage analysis on the real-world drift suite. Definitions and threshold details are in Appendix K.3.

| Dataset | Refuse (%) ↓ | Unsafe Accept (%) ↓ | Regret ↓ | Net Benefit ↑ |
|---|---|---|---|---|
| MELD | 15.2±1.8 | 2.1±0.5 | 0.4±0.1 | +0.8±0.2 |
| CMU-MOSEI | 12.5±1.5 | 2.6±0.6 | 0.5±0.1 | +0.9±0.2 |
| CC3M-500K | 18.9±2.1 | 1.7±0.4 | 0.3±0.1 | +1.2±0.3 |

## 5.8. Ablation Study and Sensitivity Analysis

Table 7 reports ablations on CMU-MOSEI and MELD under $10\times$ scaling. Removing copula stabilization causes the largest degradation increase, reaching 4.8 points on CMU-MOSEI and 4.5 points on MELD. Replacing the multiway hub coupling with pairwise alignment or replacing spectral alignment with a contrastive loss also increases degradation. These trends match the error-source decomposition: copula stabilization controls marginal sensitivity, hub coupling controls multiway coherence, and spectral learning controls shared-subspace stability.

*Table 7.* Ablation study on CMU-MOSEI and MELD under $10\times$ scaling. Degradation $\Delta$ is reported as mean $\pm$ std.

| Variant | CMU-MOSEI $\Delta \downarrow$ | MELD $\Delta \downarrow$ |
|---|---|---|
| Full SSCA | $1.8 \pm 0.1$ | $1.5 \pm 0.1$ |
| w/o Copula Stabilization | $4.8 \pm 0.3$ | $4.5 \pm 0.3$ |
| w/o Hub Coupling (use pairwise) | $3.8 \pm 0.2$ | $3.5 \pm 0.2$ |
| w/o Spectral Alignment (use Contrastive loss) | $3.5 \pm 0.2$ | $3.2 \pm 0.2$ |

Sensitivity analysis in Appendix L shows that SSCA remains stable across reasonable hyperparameter ranges: number of slicing directions $S \geq 32$, mini-batch size $m \geq 128$, stabilization weight $\lambda_{\mathrm{stab}} \in [0.001, 0.1]$, and soft-rank temperature $\tau_{\mathrm{rank}} \in [0.05, 0.2]$.

## 5.9. Computational Profile and Scaling Scope

Appendix Table 19 profiles the alignment stage under the frozen-feature protocol. Measurements exclude data loading and encoder feature extraction. At batch size 256, SSCA in SoftRank and Sketch modes reaches 22.5k and 24.8k samples/s, compared with 18.2k for InfoNCE, 18.5k for SigLIP, and 19.2k for MMML. Peak allocated memory is 4.2 GB for SoftRank and 3.9 GB for Sketch. The hard-coupling variant avoids a full multi-marginal OT solve and uses sorted one-dimensional sliced couplings with a shared hub. The spectral step uses implicit matrix-vector products with dom-

inant cost $\mathcal{O}(n_{\mathrm{iter}} d^2 mk)$ in the projected implementation; for raw-feature blocks, $k$ is replaced by the working feature dimension. These measurements support the practical claim in the studied alignment-stage setting. They do not address end-to-end foundation-model pretraining cost or encoder-side acceleration.

## 6. Limitations and Future Work

SSCA provides the strongest guarantees under approximately coordinate-wise monotone marginal distortions in feature space. Non-monotone semantic shifts, severe saturation, and heavy-tie regimes fall outside this scope. In these cases, the intended behavior is conservative fallback through gate-controlled refusal to adapt, which gives a safety policy without an out-of-scope accuracy guarantee. The coefficients in the actionable inequality and the gate thresholds are deployment-local quantities and require recalibration when the backbone, normalization pipeline, modality set, or batching policy changes. Multiway hub coupling approximates multi-marginal transport through a practical hub construction, and its behavior under very large modality counts remains an open scaling question. The missing-modality protocol in Appendix I.5 handles available modalities and standardized imputation, but a dedicated theorem for arbitrary missingness patterns is left open. The computational evidence is scoped to the frozen-feature alignment stage and does not cover end-to-end foundation-model pretraining. Broader end-to-end training, larger backbones, additional task families, non-monotone drift detectors, and long-horizon adaptation provide important directions for future work.

## 7. Conclusion

Stable Spectral Copula Alignment (SSCA) provides an auditable deployment protocol for robust multimodal alignment under a declared monotone-drift contract. SSCA controls the main error sources in approximating an ideal copula-invariant dependence operator through tie-aware copula stabilization, dependence-weighted multiway hub coupling, and diagonal-stabilized spectral learning. A calibrated, actionable Davis–Kahan inequality yields measurable subspace-risk certificates and enables a label-free stability gate for diagnostic-driven remediation within compute budgets and conservative no-update fallback outside the certified regime. The calibration factors are deployment-local rather than universal. This makes the protocol auditable under healthy unlabeled streams while keeping its formal guarantee explicitly scoped. Experiments under a frozen-feature protocol demonstrate consistent degradation reduction across controlled monotone distortions, frozen CLIP retrieval stress tests, and realistic raw-pipeline deployment drifts.

## Acknowledgements

This work was supported in part by the National Key R&D Program of China under Grant 2021 YFA0715202, Shenzhen Key Laboratory of Ubiquitous Data Enabling (Grant No. ZDSYS20220527171406015) and the Shenzhen Science and Technology Program under Grant KQTD20170810150821146 and Grant JCYJ20220530143002005.

## Impact Statement

This work studies robustness and monitoring for multimodal alignment under deployment shift. Potential benefits include more reliable multimodal systems when feature marginals change because of sensing, compression, normalization, or data-pipeline variation. The same techniques may also reduce silent failure in applications where alignment errors are difficult to detect without labels. The method does not remove the need for task-specific safety evaluation, and its stability claim is intentionally limited to the declared contract; non-monotone semantic shifts, severe quantization, and weak cross-modal dependence require conservative fallback or additional safeguards. Practitioners should combine the proposed diagnostics with domain-specific validation, privacy review, and human oversight in high-stakes settings.

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

## A. Appendix Organization and Linkage to the Main Paper

This appendix collects the technical material needed to support the main paper. It includes proof details for the stability contract, protocol definitions for calibration and gating, baseline specifications, hyperparameter settings, and additional validation results. The organization follows the structure of the main text: SSCA first defines a copula-stable contract, then derives measurable diagnostics, calibrates a deployment gate, and evaluates the resulting protocol under controlled and out-of-scope drift.

**Contents.** Appendix K states the operational contract, practical thresholds, and failure boundary. Appendix F gives the theoretical foundations and proof details. Appendix G specifies the algorithmic components, coupling implementation, and logged diagnostics. Appendices H–I provide complexity analysis, profiling details, experimental protocols, and reproducibility settings. Appendix L reports ablations, sensitivity studies, and boundary cases. The remaining sections provide additional results for retrieval, raw-pipeline drift, multi-backbone evaluation, compute-matched controls, and lightweight adapter experiments.

## B. Synthetic Ground-Truth Verification of the Actionable Proxy Bound

This section gives a controlled ground-truth check for the actionable proxy inequality in Eq. (26). The experiment directly compares the calibrated proxy bound with the true eigenspace error $\|\sin\Theta(\widehat{U}, U_*)\|_F$ under synthetic perturbations where $U_*$ is known.

A rank-$k$ symmetric operator $M_* \in \mathbb{R}^{p\times p}$ with eigengap $\gamma_*$ is first constructed. Independent perturbation sources then emulate four errors in the main decomposition: rank or soft-rank approximation error, coupling approximation error, finite-batch sampling error, and numerical stabilization error. For each randomized setting, $\widehat{U}$ is obtained from the top-$k$ eigenspace of $\widehat{M} = M_* + E$, and the true subspace error is computed from the principal angles between $\widehat{U}$ and $U_*$. The proxy right-hand side $\widehat{\mathrm{RHS}}$ uses the same diagnostic structure as Eq. (26), with the realized empirical eigengap $\widehat{\gamma}$ and the four source-specific proxy magnitudes. A scalar envelope factor $\widehat{c}_{95}$ is estimated on a held-out synthetic split and then applied to form $\widehat{\Delta} = \widehat{c}_{95}\widehat{\mathrm{RHS}}$. This calibration is used only to visualize the operational envelope and uses neither target labels nor the population eigenspace at deployment. Figure 2 reports the resulting proxy–error relation and the calibrated 95% envelope.

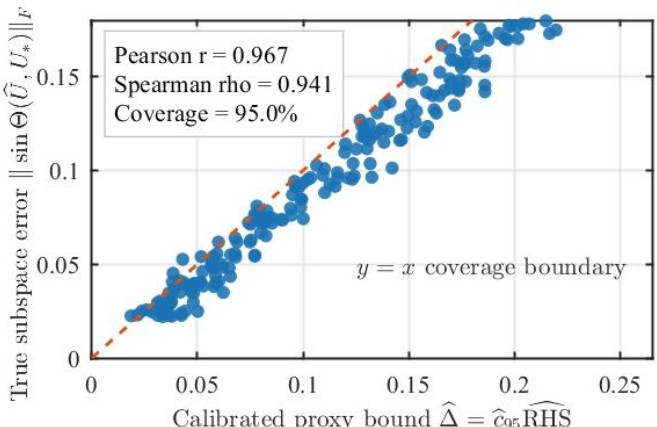

*Figure 2.* Synthetic ground-truth verification of Eq. (26): calibrated proxy bound $\widehat{\Delta} = \widehat{c}_{95}\widehat{\mathrm{RHS}}$ versus true subspace error $\|\sin\Theta(\widehat{U}, U_*)\|_F$. Across 220 randomized settings, the uncalibrated proxy exhibits strong monotone agreement with the true error (Pearson $r = 0.967$, Spearman $\rho = 0.941$), and the 95% envelope calibration achieves empirical coverage 95.0%.

## C. Baseline Closed-Loop Specification

This section specifies the baseline arms used in the closed-loop comparison. Each baseline is defined by its loss, temperature or scale parameter, positive and negative construction, symmetry, training budget, and use of memory banks or hard-negative

mining. All methods follow the frozen-feature protocol and share the same trainable projection heads $\{W^{(i)}\}$ and shared dimension $k = 128$, unless a row in Table 8 states otherwise.

## C.1. Copula-Only Substitution Controls

A key control is to separate marginal stabilization from multiway coherence and spectral identifiability. The *Copula-only* family applies the same copula stabilization step used by SSCA, then trains with a strong alignment objective under the same compute envelope.

**CopulaNorm + strong objectives.** **CopulaNorm+InfoNCE** and **CopulaNorm+SigLIP** insert the copula layer before the projection heads and use the same temperature schedules, negative construction, and symmetry settings as their corresponding baselines. These controls use the same insertion point as SSCA and isolate the effect of marginal stabilization.

**CopulaNorm + pairwise transport.** **CopulaNorm+Pairwise-SW** replaces the hub barycenter with pairwise sliced-Wasserstein alignment. Cycle repair is performed by minimizing the measured cycle defect under the same compute budget. This control tests whether hub-based multiway coupling improves coherence beyond pairwise transport with explicit cycle correction.

**Compute matching.** All substitution controls are matched to SSCA in frozen encoders, batch size, number of random projections $S$, update steps per stream step, and caching or re-encoding behavior. Each comparison reports absolute perturbed performance and degradation under the same perturbation suite, together with the corresponding CCErr and ProcRes trajectories when these diagnostics apply.

**Purpose in the closed loop.** The closed-loop specification prevents hidden degrees of freedom in baseline construction. It fixes the loss, scale or temperature, positive and negative sets, symmetry, training budget, early stopping rule, and memory-bank policy before comparing robustness and diagnostics.

## C.2. Common Training Envelope

Unless a method definition requires a different setting, all methods use the same training envelope:

Encoders are frozen; only projection heads $W^{(i)}$ and method-specific alignment parameters are trainable (Appendix E.2).

The mini-batch size $m$ follows the experiment configuration, with default $m = 256$.

Optimizer family, learning-rate schedule, validation frequency, and early-stopping patience are fixed per dataset.

Checkpoint selection and tie-breaking follow Appendix C.3.

No method uses a memory bank, queue, or hard-negative mining unless explicitly stated in Table 8.

## C.3. Checkpoint Selection and Tie-Breaking

Checkpoint selection is performed on the clean validation split of each benchmark to avoid using drift labels or drift-specific validation signals. The primary criterion is the highest clean validation score: accuracy for CMU-MOSEI and MELD, and Recall@1 for CC3M-500K. If multiple checkpoints fall within an absolute tolerance of 0.1 points, ties are broken by smaller stability diagnostics in the lexicographic order $\widehat{\varepsilon}_{\text{rank}}, \widehat{\varepsilon}_{\text{cpl}}, \widehat{\varepsilon}_{\text{samp}}, \widehat{\varepsilon}_{\text{num}}$, and then by the earliest checkpoint. This rule keeps the robustness comparison conservative because checkpoint selection does not use the drifted score.

## C.4. Canonical Loss Definitions

**Symmetric InfoNCE.** Given matched pairs $\{(u_n, v_n)\}_{n=1}^{m}$, temperature $\tau > 0$, and similarity $s(\cdot, \cdot)$, the symmetric two-modality InfoNCE loss is

$$\mathcal{L}_{\text{InfoNCE}} = \frac{1}{2m} \sum_{n=1}^{m} \left[ -\log \frac{\exp(s(u_n, v_n)/\tau)}{\sum_{j=1}^{m} \exp(s(u_n, v_j)/\tau)} - \log \frac{\exp(s(v_n, u_n)/\tau)}{\sum_{j=1}^{m} \exp(s(v_n, u_j)/\tau)} \right]. \tag{32}$$

**SigLIP-style logistic loss.** Let $y_{nj} = \mathbb{I}[n = j]$ and $\ell_{nj} = a\, s(u_n, v_j)$, where $a > 0$ is a scale parameter. The two-modality

logistic loss is

$$\mathcal{L}_{\text{SigLIP}} = \frac{1}{m^2} \sum_{n=1}^{m} \sum_{j=1}^{m} \log(1 + \exp(-(2y_{nj} - 1)\,\ell_{nj})). \tag{33}$$

When symmetric training is enabled, the same loss is also applied to the transposed pair matrix.

**Multi-view contrastive loss.** For embeddings $\{h_n^{(i)}\}$, positives are cross-modal same-sample pairs and negatives are in-batch cross-sample pairs:

$$\mathcal{L}_{\text{MV}} = \frac{1}{d(d-1)} \sum_{i \neq j} \frac{1}{m} \sum_{n=1}^{m} -\log \frac{\exp(s(h_n^{(i)}, h_n^{(j)})/\tau)}{\sum_{q=1}^{m} \exp(s(h_n^{(i)}, h_q^{(j)})/\tau)}. \tag{34}$$

**GCCA surrogate.** With centered embeddings $\bar{H}^{(i)} \in \mathbb{R}^{m \times k}$, the stochastic GCCA surrogate is

$$\mathcal{L}_{\text{GCCA}} = -\sum_{i<j} \text{tr}\left[ \left(\bar{H}^{(i)\top} \bar{H}^{(i)} + \epsilon I\right)^{-1/2} \bar{H}^{(i)\top} \bar{H}^{(j)} \left(\bar{H}^{(j)\top} \bar{H}^{(j)} + \epsilon I\right)^{-1/2} \right], \tag{35}$$

where $\epsilon > 0$ is fixed.

**Pairwise sliced-Wasserstein alignment.** Let $\{\theta_s\}_{s=1}^{S}$ be random directions in the shared embedding space. The pairwise sliced-Wasserstein alignment loss is

$$\mathcal{L}_{\text{SW}} = \sum_{i<j} \frac{1}{S} \sum_{s=1}^{S} \frac{1}{m} \sum_{t=1}^{m} \left( \langle \theta_s, h_{(t)}^{(i)} \rangle - \langle \theta_s, h_{(t)}^{(j)} \rangle \right)^2, \tag{36}$$

where $(t)$ denotes the sorted order along the projected scalar for the corresponding modality.

### C.5. Baseline Minimal Closed-Loop Table

*Table 8.* Baseline minimal closed-loop table. Each row specifies the loss form, temperature or scale, positive and negative construction, symmetry, budget and early stopping, and use of memory banks or hard-negative mining. All methods share frozen encoders, projection heads $W^{(i)}$, $k = 128$, and checkpoint selection from Appendix C.3.

| Method | Loss | Temp/scale | Positives/negatives | Sym.? | Budget/early stop | Bank/mining |
|---|---|---|---|---|---|---|
| InfoNCE | Eq. (32) | $\tau$ fixed in the experiment configuration | In-batch positives and negatives; no queue | Yes | Fixed max steps and patience | None |
| SigLIP | Eq. (33) | $a$ fixed or learnable as specified | Full $m \times m$ pair matrix; no hard mining | Yes if enabled | Same as above | None |
| 3-View Contrastive | Eq. (34) | $\tau$ fixed in the experiment configuration | Multi-positive; in-batch negatives | Yes | Same as above | None |
| GCCA | Eq. (35) | N/A | No explicit negatives; whitening regularized | N/A | Same as above | None |
| SW-Align | Eq. (36) | N/A | Sorting-based pairwise SW alignment | N/A | Same as above | None |
| SW-OT+CC | $\mathcal{L}_{\text{SW}} + \lambda_{\text{CC}}\mathcal{L}_{\text{CC}}$ | N/A | SW alignment with covariance consistency | N/A | Same as above | None |
| SSCA | Algorithm 1 | Fixed scalars in the experiment configuration | Hub-based multiway coupling; no contrastive negatives required | N/A | Same as above | None |
| SSCA+Contrastive | SSCA + Eq. (32) | $\tau$ fixed in the experiment configuration | Adds in-batch negatives on stabilized embeddings | Yes | Same as above | None |

**Fairness closure.** Table 8 closes the baseline specification along four axes: asymmetric training choices, negative construction, hyperparameter selection, and matched update budgets or checkpoint rules.

## D. Streaming Sensitivity and Long-Horizon Protocols

**Closed-loop stability under horizon variation.** This section specifies how the streaming diagnostics are checked across horizon lengths. The goal is to verify that the observed diagnostic trends are caused by structural drift rather than by a short burn-in window or a transient adaptation artifact.

**Protocol definition.** Each streaming experiment uses the same ordered stream, drift injection point, update rule, and hyperparameter setting. Only the effective horizon length changes. Two regimes are considered: short-horizon updates with $T = 1$–$5$ passes over the data, and long-horizon deployment with approximately $T = 6000$ streaming batches. The gate, remediation rule, and no-update fallback are applied identically in both regimes.

**Interpretation.** Stable diagnostic ordering across short and long horizons indicates that the proxy terms localize persistent contract violations. Large horizon-dependent changes indicate that the batch window or adaptation dynamics dominate the measured signal. This protocol links the deployment stream to the same diagnostic quantities used in the main gate, without introducing additional selection criteria.

## E. Notation and Additional Definitions

### E.1. Core Symbols

$d$ denotes the number of modalities, and $i, j \in [d]$ index modalities. $p_i$ is the raw feature dimension of modality $i$, and $k$ is the shared embedding dimension, fixed to $128$ unless otherwise stated. $m$ is the mini-batch size, and $t$ is the streaming batch index.

For modality $i$, $x^{(i)} \in \mathbb{R}^{p_i}$ denotes a frozen encoder feature and $H^{(i)} \in \mathbb{R}^{m \times p_i}$ denotes the corresponding batch feature matrix. The stabilized feature matrix is $G^{(i)} = \mathcal{T}^{(i)}(H^{(i)}) \in \mathbb{R}^{m \times p_i}$, where $\mathcal{T}^{(i)}$ denotes the clipped rank-probit stabilization map. The trainable projection is $W^{(i)} \in \mathbb{R}^{p_i \times k}$, and the projected batch embedding is $Z_0^{(i)} = G^{(i)} W^{(i)} \in \mathbb{R}^{m \times k}$. The row vector $h_n^{(i)} \in \mathbb{R}^k$ denotes the embedding of sample $n$ in modality $i$.

To avoid overloading the operator symbol $M$, modality availability is denoted by $\mathbf{a} \in \{0, 1\}^d$, where $a_i = 1$ means that modality $i$ is observed. $\omega_{ij}$ denotes the dependence weight between modalities $i$ and $j$, and $w_i$ denotes the modality weight used in the hub barycenter. $\lambda_{\text{stab}}$ is the diagonal stabilization parameter for the spectral operator. $\alpha_{\text{clip}}$ (written as $\alpha$ in the main text when unambiguous) is the clipping level for the probit map.

**Latent rank versus embedding dimension.** The latent factor rank used in identifiability arguments is denoted by $r$. It is distinct from the shared embedding dimension $k$.

### E.2. Trainable Parameter Set

All encoders are frozen. The trainable parameter set is

$$\Theta_{\text{train}} \triangleq \{W^{(i)}\}_{i=1}^d \cup \Theta_{\text{align}}, \tag{37}$$

where $\Theta_{\text{align}}$ contains learned alignment scalars when they are enabled. Fixed temperatures, clipping values, sketch sizes, and stabilization constants are treated as hyperparameters and are excluded from the optimization state. Any ablation that introduces trainable normalizers includes them in $\Theta_{\text{align}}$ and reports their parameter counts.

### E.3. Normalized Rank and Soft-Rank Operators

For a batch vector $v \in \mathbb{R}^m$, let $R_\ell(v)$ be the deterministic rank assigned to entry $v_\ell$ after the tie-handling rule in Appendix G.1. The normalized rank is

$$\text{uRank}(v)_\ell \triangleq \frac{R_\ell(v) - \frac{1}{2}}{m} \in (0, 1). \tag{38}$$

For tied blocks, the same deterministic tie policy is used throughout the batch, and the corresponding tie mass is logged as a diagnostic quantity. The differentiable approximation $\text{sRank}_\tau(v)$ uses the soft-sorting operator with temperature $\tau = \tau_{\text{rank}}$ and returns values in $(0, 1)$ that approximate $\text{uRank}(v)$; details appear in Appendix F.4.

# F. Theoretical Foundations and Proofs

This section provides the proofs and auxiliary results used by the stability contract in the main paper. The results separate three levels of claims: exact order invariance under orientation-preserving monotone transforms, finite-sample perturbation control through rank and transport proxies, and eigenspace stability through a Davis–Kahan argument. All statements use the notation in Appendix E. The formal invariance claim applies to approximately coordinate-wise increasing marginal distortions in feature space; non-monotone distortions and severe tie regimes are treated by the diagnostics and fallback protocol.

## F.1. Order Preservation Under Monotone Transforms

**Lemma F.1** (Rank behavior under strictly monotone transforms)*. Let $g : \mathbb{R} \to \mathbb{R}$ be strictly monotone. For any batch vector $v \in \mathbb{R}^m$ without ties, the normalized rank satisfies*

$$\text{uRank}(g(v))_\ell = \begin{cases} \text{uRank}(v)_\ell, & g \text{ is strictly increasing,} \\ 1 - \text{uRank}(v)_\ell, & g \text{ is strictly decreasing.} \end{cases} \tag{39}$$

*Proof.* Strictly increasing maps preserve the total order of all entries. Thus the sorting permutation is unchanged, and the normalized rank is unchanged. Strictly decreasing maps reverse the total order. If entry $v_\ell$ has rank $r_\ell \in \{1, \ldots, m\}$, then $g(v_\ell)$ has rank $m + 1 - r_\ell$. By the definition

$$\text{uRank}(v)_\ell = \frac{r_\ell - \frac{1}{2}}{m},$$

the reversed rank satisfies

$$\frac{m + 1 - r_\ell - \frac{1}{2}}{m} = 1 - \frac{r_\ell - \frac{1}{2}}{m}.$$

This proves Eq. (39). ∎

**Operational scope.** The SSCA stability contract uses the orientation-preserving case. Strictly decreasing transformations fall outside the formal copula-stability contract unless the orientation reversal is explicitly detected and corrected. This distinction prevents rank reversal from being counted as a stable marginal distortion.

## F.2. Kendall's $\tau$ Under Monotone Transforms

**Lemma F.2** (Monotone behavior of Kendall's $\tau$)*. Let $(X, Y)$ be real-valued random variables with a continuous joint distribution. For any strictly increasing functions $g$ and $h$,*

$$\tau(X, Y) = \tau(g(X), h(Y)). \tag{40}$$

*If exactly one of $g$ and $h$ is strictly decreasing, then*

$$\tau(g(X), h(Y)) = -\tau(X, Y). \tag{41}$$

*If both are strictly decreasing, the sign is preserved.*

*Proof.* Kendall's $\tau$ depends on the signs of pairwise products

$$(X - X')(Y - Y'),$$

where $(X', Y')$ is an independent copy of $(X, Y)$. Strictly increasing maps preserve the sign of each one-dimensional difference. A strictly decreasing map flips the sign of the corresponding difference. Therefore, applying two increasing maps preserves concordance and discordance, applying exactly one decreasing map swaps them, and applying two decreasing maps preserves their product sign. The stated identities follow from the definition of Kendall's $\tau$. ∎

**Operational implication.** Lemma F.2 justifies using Kendall-type dependence estimates for the weights $\omega_{ij}$ under orientation-preserving monotone marginal distortions. Finite-sample error and ties are handled by the diagnostics in Appendix G.4 and by the tie policy in Appendix G.1.

### F.3. Proof of Theorem 3.1: Finite-Sample Order Stability

*Proof.* Consider one coordinate and one streaming batch $v = (v_1, \ldots, v_m) \in \mathbb{R}^m$. The empirical CDF is

$$F_t(x) = \frac{1}{m} \sum_{j=1}^{m} \mathbb{I}\{v_j \le x\}. \tag{42}$$

Let $\widetilde{F}_t$ be the CDF estimate produced by the sketch or soft-rank routine. The clipped rank–probit map is

$$\phi_t(x) = \Phi^{-1}\Big(\text{clip}\Big(\widetilde{F}_t(x), \alpha_{\text{clip}}, 1 - \alpha_{\text{clip}}\Big)\Big). \tag{43}$$

Let $g$ be strictly increasing and write $v' = g(v)$. Denote the transformed empirical CDF and its estimate by $F_t^{(g)}$ and $\widetilde{F}_t^{(g)}$.

**Step 1: CDF estimation error.** Assume the CDF estimator has batch-wise uniform error

$$\varepsilon_t \triangleq \sup_{z \in \mathbb{R}} |\widetilde{F}_t(z) - F_t(z)|. \tag{44}$$

The same estimator class applied to the transformed batch satisfies $\sup_{z \in \mathbb{R}} |\widetilde{F}_t^{(g)}(z) - F_t^{(g)}(z)| \le \varepsilon_t$.

**Step 2: Order-preserving empirical CDF relation.** For a strictly increasing $g$, the empirical CDFs satisfy $F_t^{(g)}(g(x)) = F_t(x)$ at all continuity points of the empirical rank map. With deterministic tie handling, the implemented rank value can differ from the ideal no-tie rank by at most the maximum tie mass

$$\delta'_{\text{tie}} \triangleq \max_u \frac{\#\{j : v_j = u\}}{m}. \tag{45}$$

**Step 3: CDF discrepancy bound.** For any batch point $x$,

$$\left|\widetilde{F}_t^{(g)}(g(x)) - \widetilde{F}_t(x)\right| \le \left|\widetilde{F}_t^{(g)}(g(x)) - F_t^{(g)}(g(x))\right| + \left|F_t^{(g)}(g(x)) - F_t(x)\right| + \left|F_t(x) - \widetilde{F}_t(x)\right|$$

$$\le 2\varepsilon_t + \delta'_{\text{tie}}. \tag{46}$$

**Step 4: Propagation through clipping and probit.** The clipping map is 1-Lipschitz. On $[\alpha_{\text{clip}}, 1 - \alpha_{\text{clip}}]$, the inverse Gaussian CDF is Lipschitz with constant

$$L_{\Phi^{-1}}(\alpha_{\text{clip}}) = \sup_{u \in [\alpha_{\text{clip}}, 1 - \alpha_{\text{clip}}]} \left|\frac{d}{du} \Phi^{-1}(u)\right| = \frac{1}{\varphi(\Phi^{-1}(\alpha_{\text{clip}}))}, \tag{47}$$

where $\varphi$ is the standard normal density. Thus

$$\left|\phi_t^{(g)}(g(x)) - \phi_t(x)\right| \le L_{\Phi^{-1}}(\alpha_{\text{clip}}) \left(2\varepsilon_t + \delta'_{\text{tie}}\right). \tag{48}$$

Setting $\delta_{\text{tie}} \triangleq L_{\Phi^{-1}}(\alpha_{\text{clip}})\delta'_{\text{tie}}$ gives the finite-sample order-stability bound.

**Numerical scale.** For $\alpha_{\text{clip}} = 0.005$, $L_{\Phi^{-1}}(0.005) \approx 69.0$. This explains why SSCA logs rank approximation and tie-mass diagnostics: small CDF errors can be amplified near the clipped tails.

### F.4. Quantile Sketch and Soft-Rank Approximation

**Proposition F.3** (Sketch-mode CDF proxy). *Assume that each coordinate maintains a quantile sketch whose batch-wise CDF error satisfies, with probability at least $1 - \delta$,*

$$\sup_{z \in \mathbb{R}} |\widetilde{F}_t(z) - F_t(z)| \le \varepsilon_{\text{sketch}}(s_{\text{sketch}}, \delta). \tag{49}$$

*If the streaming update introduces an additional discretization error bounded by $\varepsilon_{\text{upd}}$, then the rank-side contribution used in Eq. (48) can be bounded by $\varepsilon_t \le \varepsilon_{\text{sketch}}(s_{\text{sketch}}, \delta) + \varepsilon_{\text{upd}}$.*

*Proof.* The statement follows by the triangle inequality between the empirical CDF, the maintained sketch CDF, and the incrementally updated sketch state. The first term is the sketch approximation error; the second term accounts for update discretization in streaming operation.

**Proposition F.4** (Soft-rank approximation proxy). *Let* $\mathrm{sRank}_\tau$ *be the differentiable soft-rank operator used during training. Define the batch-grid discrepancy*

$$\widehat{\varepsilon}_{\mathrm{rank}} \triangleq \max_{\ell \in [m]} |\mathrm{sRank}_\tau(v)_\ell - \mathrm{uRank}(v)_\ell| . \tag{50}$$

*Then the finite-sample order-stability bound in Eq. (48) holds on the batch grid with $\varepsilon_t$ replaced by $\widehat{\varepsilon}_{\mathrm{rank}}$, up to floating-point precision.*

*Proof.* The quantity $\widehat{\varepsilon}_{\mathrm{rank}}$ is the maximum pointwise deviation between the differentiable rank approximation and the deterministic normalized rank over the observed batch. Substituting this batch-grid deviation into Eq. (46) yields the same Lipschitz propagation bound after clipping and probit transformation.

### F.5. Sliced-Wasserstein Monte Carlo Approximation

**Proposition F.5** (Finite-$S$ concentration). *Let* $\{\theta_s\}_{s=1}^S$ *be i.i.d. directions drawn uniformly from* $\mathbb{S}^{k-1}$. *For fixed empirical measures $P$ and $Q$, define $Z_s = \mathcal{W}_2^2(P_{\theta_s}, Q_{\theta_s})$. If $0 \le Z_s \le B$ almost surely, then for any $\delta \in (0,1)$,*

$$\left| \frac{1}{S} \sum_{s=1}^S Z_s - \mathbb{E}_\theta Z_\theta \right| \le B\sqrt{\frac{\log(2/\delta)}{2S}} \quad \text{with probability at least } 1 - \delta. \tag{51}$$

*Proof.* The result is Hoeffding's inequality applied to the bounded random variables $Z_s$. The expectation $\mathbb{E}_\theta Z_\theta$ is the sliced-Wasserstein objective over random one-dimensional projections.

### F.6. Shared-Subspace Identifiability and Eigengap Separation

**Proposition F.6** (Shared-subspace separation). *Consider the centered multi-view factor model*

$$\bar{G}^{(i)} = SA_i^\top + N^{(i)}, \qquad i \in [d], \tag{52}$$

*where $S \in \mathbb{R}^{m \times r}$, $m^{-1}\mathbb{E}[S^\top S] = \Sigma_S \succ 0$, and each $A_i \in \mathbb{R}^{p_i \times r}$ has full column rank. Assume cross-view noise is uncorrelated with the shared factor and with other views. Let $\mathcal{B}_*$ be the population signal block operator with off-diagonal blocks*

$$(\mathcal{B}_*)_{ij} = \omega_{ij} A_i \Sigma_S A_j^\top, \qquad i \ne j, \tag{53}$$

*and zero diagonal blocks. If $\mathcal{B}_*$ has a $k$-dimensional dominant eigenspace separated by an eigengap $\gamma_* > 0$, then this eigenspace is identifiable from population cross-view dependence. The stabilized operator*

$$\mathcal{M}_* = \mathcal{B}_* + \lambda_{\mathrm{stab}} I \tag{54}$$

*has the same eigenspaces and the same eigengap at rank $k$.*

*Proof.* The cross-covariance between views $i$ and $j$ is

$$C_{ij,*} = \mathbb{E}\left[ m^{-1} \bar{G}^{(i)\top} \bar{G}^{(j)} \right] = A_i \Sigma_S A_j^\top, \qquad i \ne j, \tag{55}$$

because cross-view noise terms vanish by assumption. Thus all off-diagonal blocks of $\mathcal{B}_*$ are functions of the same shared factor covariance $\Sigma_S$ and the view maps $\{A_i\}$. The dominant eigenspace of $\mathcal{B}_*$ therefore lies in the span of the block view maps and represents the shared cross-view signal. The assumed eigengap $\gamma_* > 0$ makes this shared subspace identifiable as an isolated invariant subspace. Adding $\lambda_{\mathrm{stab}} I$ shifts all eigenvalues by the same amount and leaves eigenvectors and eigengaps unchanged. Hence $\mathcal{M}_*$ and $\mathcal{B}_*$ share the same identifiable top-$k$ eigenspace.

**A sufficient separation condition.** Let $\widetilde{A} = \mathrm{blkdiag}(A_1, \ldots, A_d)$ and let $\Omega$ be the symmetric modality-weight matrix with zero diagonal and off-diagonal entries $\omega_{ij}$. Then

$$\mathcal{B}_* = \widetilde{A}(\Omega \otimes \Sigma_S)\widetilde{A}^\top. \tag{56}$$

If $\underline{a}I \preceq A_i^\top A_i \preceq \overline{a}I$ for all $i$, and the positive part of $\Omega$ has a dominant eigenvalue separated from the next positive mode, then a sufficient lower bound is

$$\gamma_* \geq \underline{a}\,\lambda_1^+(\Omega)\,\sigma_r(\Sigma_S) - \overline{a}\,\lambda_2^+(\Omega)\,\sigma_1(\Sigma_S), \tag{57}$$

whenever the right-hand side is positive. Here $\lambda_1^+(\Omega)$ and $\lambda_2^+(\Omega)$ denote the largest and second-largest positive eigenvalues of $\Omega$.

## F.7. Hub Quantile as a One-Dimensional Wasserstein-2 Barycenter

For a fixed direction $\theta \in \mathbb{S}^{k-1}$, let $\widehat{Q}_\theta^{(i)}(u)$ be the empirical quantile function of $\langle \theta, Z_0^{(i)} \rangle$. The one-dimensional Wasserstein-2 barycenter quantile under weights $\{w_i\}_{i=1}^d$ is

$$Q_\theta^{(B)}(u) = \sum_{i=1}^d w_i \widehat{Q}_\theta^{(i)}(u), \tag{58}$$

because in one dimension $\mathcal{W}_2^2$ equals the squared $L^2$ distance between quantile functions, and the minimizer of a weighted sum of squared distances is their weighted average.

**Lemma F.7** (One-dimensional optimal transport sorting and hub surrogate)**.** *Let $\mu = \frac{1}{m}\sum_{\ell=1}^m \delta_{a_\ell}$ and $\nu = \frac{1}{m}\sum_{\ell=1}^m \delta_{b_\ell}$ be empirical measures on $\mathbb{R}$ with equal weights. Let $a_{(1)} \leq \cdots \leq a_{(m)}$ and $b_{(1)} \leq \cdots \leq b_{(m)}$ denote sorted samples. Then*

$$\mathcal{W}_2^2(\mu, \nu) = \frac{1}{m}\sum_{\ell=1}^m \left(a_{(\ell)} - b_{(\ell)}\right)^2. \tag{59}$$

*For empirical measures $\mu_i = \frac{1}{m}\sum_{\ell=1}^m \delta_{a_\ell^{(i)}}$ and weights $w_i \geq 0$ with $\sum_i w_i = 1$, a one-dimensional barycenter is*

$$\nu^B = \frac{1}{m}\sum_{\ell=1}^m \delta_{q_{(\ell)}}, \qquad q_{(\ell)} = \sum_{i=1}^d w_i a_{(\ell)}^{(i)}. \tag{60}$$

*Proof.* In one dimension, the monotone rearrangement is optimal for quadratic transport. Thus the optimal plan pairs the $\ell$-th order statistic of $\mu$ with the $\ell$-th order statistic of $\nu$, which gives Eq. (59). The barycenter objective is

$$\sum_{i=1}^d w_i \frac{1}{m}\sum_{\ell=1}^m \left(a_{(\ell)}^{(i)} - q_{(\ell)}\right)^2.$$

It separates over $\ell$. Minimizing each scalar quadratic gives $q_{(\ell)} = \sum_i w_i a_{(\ell)}^{(i)}$, proving Eq. (60). $\qquad \blacksquare$

## F.8. Proof of Lemma 3.2: Equivalence to a Top-$k$ Eigenspace Problem

*Proof.* Consider the whitened block variables $U_i$ and the stacked matrix $U = [U_1^\top, \ldots, U_d^\top]^\top, \quad U^\top U = I_k$. Let $\mathcal{M}$ be the symmetric block operator with off-diagonal blocks $\mathcal{M}_{ij} = \omega_{ij} R_i C_{ij} R_j, \quad i \neq j$, and zero diagonal blocks. Then

$$\mathrm{tr}(U^\top \mathcal{M} U) = 2 \sum_{1 \leq i < j \leq d} \omega_{ij}\,\mathrm{tr}(U_i^\top R_i C_{ij} R_j U_j). \tag{61}$$

Under the reparameterization $W_i = R_i U_i$, the right-hand side is the symmetrized form of the weighted cross-view trace objective in Eq. (21). By the Ky Fan theorem, the relaxed block-trace objective is optimized by the top-$k$ eigenspace of $\mathcal{M}$. $\qquad \blacksquare$

### F.9. Eigenspace Perturbation via Davis–Kahan

**Theorem F.8** (Eigenspace stability). *Let $\mathcal{M}_*$ be a symmetric population operator with eigengap $\gamma_* = \lambda_k(\mathcal{M}_*) - \lambda_{k+1}(\mathcal{M}_*) > 0$. Let $\widehat{\mathcal{M}} = \mathcal{M}_* + E$ be an empirical estimate, and let $U_*$ and $\widehat{U}$ be orthonormal bases of the corresponding top-$k$ eigenspaces. If $\|E\|_2 \leq \gamma_*/2$, then*

$$\|\sin\Theta(\widehat{U}, U_*)\|_2 \leq \frac{2\|E\|_2}{\gamma_*}, \tag{62}$$

*and*

$$\|\sin\Theta(\widehat{U}, U_*)\|_F \leq \frac{2\sqrt{k}\|E\|_2}{\gamma_*}. \tag{63}$$

*Proof.* The spectral-norm inequality follows from the Davis–Kahan $\sin\Theta$ theorem under an eigengap $\gamma_*$ and perturbation size below the gap scale. The Frobenius bound follows from $\|\sin\Theta(\widehat{U}, U_*)\|_F \leq \sqrt{k}\|\sin\Theta(\widehat{U}, U_*)\|_2$.

### F.10. From Operator Perturbation to the Actionable Proxy Bound

**Corollary F.9** (Proxy-controlled subspace deviation). *Assume the empirical block-operator perturbation admits the decomposition*

$$E = E_{\text{rank}} + E_{\text{cpl}} + E_{\text{samp}} + E_{\text{num}}, \tag{64}$$

*and the operator norm of each component is controlled on the calibration window by*

$$\|E_{\text{rank}}\|_2 + \|E_{\text{cpl}}\|_2 + \|E_{\text{samp}}\|_2 + \|E_{\text{num}}\|_2 \leq c_{\text{rank}}\widehat{\varepsilon}_{\text{rank}} + c_{\text{cpl}}\widehat{\varepsilon}_{\text{cpl}} + c_{\text{samp}}\widehat{\varepsilon}_{\text{samp}} + c_{\text{num}}\widehat{\varepsilon}_{\text{num}}. \tag{65}$$

*If $\widehat{\gamma}$ is the empirical eigengap used by the gate and remains in the stable regime, then the operational proxy bound takes the form*

$$\|\sin\Theta(\widehat{U}, U_*)\|_F \lesssim \frac{c_{\text{rank}}\widehat{\varepsilon}_{\text{rank}} + c_{\text{cpl}}\widehat{\varepsilon}_{\text{cpl}} + c_{\text{samp}}\widehat{\varepsilon}_{\text{samp}} + c_{\text{num}}\widehat{\varepsilon}_{\text{num}}}{\widehat{\gamma}}. \tag{66}$$

*Proof.* By the triangle inequality,

$$\|E\|_2 \leq \|E_{\text{rank}}\|_2 + \|E_{\text{cpl}}\|_2 + \|E_{\text{samp}}\|_2 + \|E_{\text{num}}\|_2.$$

Substituting Eq. (65) into the Davis–Kahan bound in Eq. (63) gives the proxy-controlled subspace deviation. The constants $c_{\text{rank}}, c_{\text{cpl}}, c_{\text{samp}}, c_{\text{num}}$ are deployment-local calibration factors, and $\widehat{\gamma}$ is used as the observable eigengap in the gate. This yields Eq. (66).

**Interpretation.** The corollary gives the theoretical form behind the actionable inequality used in the main text. The coefficients are calibrated from healthy unlabeled batches and should be recalibrated when the backbone, normalization pipeline, modality set, or batching policy changes. When the calibrated proxy or the eigengap leaves the stable regime, the gate rejects the update and the protocol uses the no-update fallback.

## G. Algorithmic Details and Implementation

This section specifies the mini-batch operations, coupling construction, diagnostic quantities, and coherence metrics used by SSCA. The notation follows Appendix E: $H^{(i)}$ denotes frozen encoder features, $G^{(i)} = \mathcal{T}^{(i)}(H^{(i)})$ denotes stabilized features, $W^{(i)}$ denotes the trainable projection, and $Z_0^{(i)} = G^{(i)}W^{(i)}$ denotes projected embeddings before coupling.

### G.1. Deterministic Tie-Breaking and Quantization

All sorting-based operations use stable sorting by the original sample index. This includes soft-rank calibration, one-dimensional OT sorting, and Kendall-type dependence estimation. For $v \in \mathbb{R}^m$, the sorting permutation $\sigma$ satisfies $v_{\sigma(1)} \leq \cdots \leq v_{\sigma(m)}$. If $v_a = v_b$ and $a < b$, then the stable sorting rule enforces $\sigma^{-1}(a) < \sigma^{-1}(b)$. The maximum tie mass is recorded for each batch:

$$\text{TieMass}(v) \triangleq \max_u \frac{\#\{j : v_j = u\}}{m}. \tag{67}$$

The order-stability bound uses this value through Eq. (45). Thus the finite-sample analysis avoids an implicit tie-free assumption. When heavy quantization or saturation produces large tie blocks, $\mathrm{TieMass}(v)$ becomes a direct diagnostic of the boundary where rank-based stabilization is no longer reliable.

## G.2. Distortion and Drift Application Operator

All feature-space distortions are applied to the representation that enters SSCA before stabilization, coupling, and fusion or scoring. In this subsection, $h^{(i)}$ denotes the task-level representation for modality $i$, and $\mathbf{a} \in \{0,1\}^d$ denotes the modality-availability mask, where $a_i = 1$ means that modality $i$ is present.

**Classification tasks.** For monotone-distortion robustness, the feature-space transform is applied as

$$h^{(i)} \leftarrow \phi(h^{(i)}), \qquad \forall i \text{ such that } a_i = 1. \tag{68}$$

The downstream classifier uses available-only fusion,

$$\bar{h} = \frac{\sum_{i=1}^d a_i h^{(i)}}{\sum_{i=1}^d a_i}, \tag{69}$$

followed by a linear prediction head.

**Retrieval tasks.** Retrieval uses normalized image and text embeddings:

$$s = \left\langle \frac{h^{(\mathrm{img})}}{\|h^{(\mathrm{img})}\|_2}, \frac{h^{(\mathrm{txt})}}{\|h^{(\mathrm{txt})}\|_2} \right\rangle. \tag{70}$$

Affine scaling cancels under this normalization. The retrieval robustness study therefore uses non-affine monotone feature distortions before normalization:

$$\widehat{h} \leftarrow \frac{\phi(h)}{\|\phi(h)\|_2}. \tag{71}$$

**Streaming drift adaptation.** Abrupt streaming drift is injected by scaling the video-modality representation:

$$h^{(v)} \leftarrow 10\, h^{(v)}. \tag{72}$$

This stress test changes the feature marginal while preserving the sample order within most coordinates, matching the in-scope monotone-drift setting.

## G.3. Hub Coupling: From One-Dimensional OT to a Batch-Level Coupling

This subsection defines the batch-level coupling used by the hub alignment step. The coupling is computed in the projected shared space $Z_0^{(i)}$, then applied to the stabilized feature matrix $G^{(i)}$ before the spectral update.

**Per-direction OT permutation matrices.** Fix a direction $\theta_s \in \mathbb{S}^{k-1}$. Let $p_s^{(i)}(t) = \langle \theta_s, Z_0^{(i)}(t) \rangle$, $t = 1, \ldots, m$, be the projected scalar values for modality $i$. Let $\sigma_s^{(i)}$ be the stable sorting permutation of $p_s^{(i)}$. The one-dimensional OT map between two empirical distributions matches sorted samples. The direction-specific permutation from modality $i$ to modality $j$ is

$$\Pi_s^{(i \to j)} \triangleq P(\sigma_s^{(j)}) P(\sigma_s^{(i)})^\top \in \{0,1\}^{m \times m}, \tag{73}$$

where $P(\sigma)$ is the permutation matrix induced by $\sigma$.

**Hub selection.** For each modality $i$, define the hub discrepancy score

$$\beta_i \triangleq \frac{1}{S} \sum_{s=1}^S \frac{1}{m} \sum_{t=1}^m \left( p_{s,(t)}^{(i)} - Q_{\theta_s}^{(B)}(u_t) \right)^2, \qquad u_t = \frac{t - \frac{1}{2}}{m}, \tag{74}$$

where $p_{s,(t)}^{(i)}$ is the $t$-th order statistic and $Q_{\theta_s}^{(B)}$ is the one-dimensional barycenter quantile from Appendix F.7. The hub modality is

$$i^\star = \arg\min_i \beta_i, \tag{75}$$

and the hub coupling is set to the identity.

**Soft coupling.** For each $i \neq i^\star$, the direction-specific OT permutations are averaged:

$$\widetilde{P}_i \triangleq \frac{1}{S} \sum_{s=1}^{S} \Pi_s^{(i \to i^\star)}. \tag{76}$$

A fixed Sinkhorn normalization maps $\widetilde{P}_i$ to a doubly stochastic matrix:

$$P_i \triangleq \text{Sinkhorn}(\widetilde{P}_i; \tau_{\text{sk}}, n_{\text{sk}}), \tag{77}$$

where $\tau_{\text{sk}}$ and $n_{\text{sk}}$ are fixed in the experiment configuration.

**Hard coupling.** Hard coupling uses the same averaged plan $\widetilde{P}_i$ and rounds it to a permutation matrix:

$$\Pi_i \triangleq \arg \max_{\Pi \in \mathcal{P}_m} \langle \Pi, \widetilde{P}_i \rangle, \tag{78}$$

where $\mathcal{P}_m$ is the set of $m \times m$ permutation matrices. This assignment is computed by Hungarian matching. Hard coupling enforces an exact permutation structure. This alone does not certify that the estimated transport plan is accurate. The coupling quality is therefore audited by the rounding and hub-margin proxies in Appendix G.4.

**Applying the coupling.** The batch coupling is applied to stabilized features:

$$\widetilde{G}^{(i)} = \begin{cases} P_i G^{(i)}, & \text{soft coupling}, \\ \Pi_i G^{(i)}, & \text{hard coupling}, \end{cases} \tag{79}$$

followed by centering and spectral learning.

## G.4. Diagnostics

This subsection defines the diagnostic proxies used by the closed-loop protocol. Each quantity is computed per batch. The eigengap-normalized versions are used in the operational proxy bound and gate.

**Rank-calibration proxy.**

$$\widehat{\varepsilon}_{\text{rank}} \triangleq \max_{\text{coord}} \max_{\ell \in [m]} |\text{sRank}_{\tau_{\text{rank}}}(v)_\ell - \text{uRank}(v)_\ell|. \tag{80}$$

**CDF/quantile estimation proxy.** When sketch mode is used, the quantile proxy is

$$\widehat{\varepsilon}_{\text{qnt}} \triangleq \max_{\text{coord}} \max_{u \in \mathcal{U}} \left| \widetilde{Q}(u) - \widehat{Q}(u) \right|, \qquad \mathcal{U} = \left\{ \frac{t - \frac{1}{2}}{m} \right\}_{t=1}^{m}, \tag{81}$$

where $\widehat{Q}$ is the empirical batch quantile and $\widetilde{Q}$ is the sketch-derived quantile on the same grid. This proxy is zero in pure soft-rank mode.

**Soft-coupling doubly stochastic defect.** For soft coupling,

$$\widehat{\varepsilon}_{\text{ds}} \triangleq \max_i \left( \|P_i \mathbf{1} - \mathbf{1}\|_\infty + \|P_i^\top \mathbf{1} - \mathbf{1}\|_\infty \right). \tag{82}$$

This value is zero for exact hard permutations, so the hard-coupling quality proxy uses the rounding and hub diagnostics.

**Rounding and directional disagreement.** For hard coupling and soft-coupling audits, define

$$\widehat{\varepsilon}_{\text{rnd}} \triangleq \max_i \frac{\|\Pi_i - \widetilde{P}_i\|_F}{\sqrt{m}}. \tag{83}$$

For hard coupling, $\Pi_i$ is the permutation used by the method. For soft coupling, $\Pi_i$ is the Hungarian rounding of $\widetilde{P}_i$ used only for auditing. A large value indicates disagreement among one-dimensional OT permutations across slicing directions.

**Hub identifiability margin.** The hub-margin diagnostic is

$$\widehat{\varepsilon}_{\text{hub}} \triangleq \frac{\beta_{(2)} - \beta_{(1)}}{\beta_{(1)} + \epsilon_{\text{hub}}}, \qquad \epsilon_{\text{hub}} = 10^{-8}, \tag{84}$$

where $\beta_{(1)} \leq \beta_{(2)}$ are the two smallest values among $\{\beta_i\}_{i=1}^d$. A small margin indicates hub ambiguity and may coincide with unstable couplings.

**Coupling proxy used by the gate.** The coupling-side proxy in the main diagnostic dictionary is computed from the relevant coupling diagnostics:

$$\widehat{\varepsilon}_{\text{cpl}} \triangleq \max\left\{\widehat{\varepsilon}_{\text{rnd}}, \frac{1}{\widehat{\varepsilon}_{\text{hub}} + \epsilon_{\text{hub}}}, \widehat{\varepsilon}_{\text{ds}}\right\}, \tag{85}$$

with $\widehat{\varepsilon}_{\text{ds}} = 0$ under hard coupling. This definition maps directional disagreement, hub ambiguity, and soft-coupling normalization defects into a single coupling-risk proxy.

**Sample-noise proxy.**

$$\widehat{\varepsilon}_{\text{samp}} \triangleq \max_{i \neq j} \left\|C_{ij} - C_{ij}^{(\text{EMA})}\right\|_F, \tag{86}$$

where $C_{ij}^{(\text{EMA})}$ is an exponential-moving-average reference of the cross-covariance.

**Numerical residual.**

$$\widehat{\varepsilon}_{\text{num}} \triangleq \frac{\|\mathcal{M}\widehat{U} - \widehat{U}\Lambda\|_F}{\|\widehat{U}\|_F}. \tag{87}$$

**Eigengap estimate.** For each batch, the eigensolver returns leading eigenvalues $\{\widehat{\lambda}_\ell\}$. The empirical eigengap is

$$\widehat{\gamma}(t) \triangleq \widehat{\lambda}_k(t) - \widehat{\lambda}_{k+1}(t). \tag{88}$$

The protocol also records $\widehat{\varepsilon}_\bullet(t)/(\widehat{\gamma}(t) + 10^{-8})$ for the proxy terms used by the operational bound.

### G.5. Coherence Metrics

This subsection defines ProcRes and CCErr, which are used for cycle auditing and stability monitoring. Both are computed in the shared projected space after coupling.

*Table 9.* Coherence metrics used for cycle auditing and stability monitoring. Lower is better.

| Metric | Exact definition | Use |
|--------|------------------|-----|
| ProcRes | Eq. (90) | Procrustes coherence |
| CCErr | Eq. (91) | Rank-cycle coherence |

**ProcRes.** Let $\widetilde{Z}^{(i)} \triangleq \widetilde{G}^{(i)} W^{(i)} \in \mathbb{R}^{m \times k}$ be the coupled projected embedding. Let $\bar{Z}^{(i)}$ denote its centered version. For the hub modality $i^\star$, define

$$R_i \triangleq \arg\min_{R^\top R = I_k} \left\|\bar{Z}^{(i)} - \bar{Z}^{(i^\star)} R\right\|_F. \tag{89}$$

The solution is given by the orthogonal Procrustes SVD. The residual is

$$\text{ProcRes} \triangleq \frac{1}{d-1} \sum_{i \neq i^\star} \frac{\left\|\bar{Z}^{(i)} - \bar{Z}^{(i^\star)} R_i\right\|_F}{\left\|\bar{Z}^{(i)}\right\|_F + 10^{-8}}. \tag{90}$$

**CCErr.** Let $U^{(i)} \in (0,1)^{m \times k}$ denote coordinate-wise normalized ranks computed on the coupled projected embedding $\widetilde{Z}^{(i)}$: $U_{t,c}^{(i)} = \text{uRank}\left([\widetilde{Z}_{1,c}^{(i)}, \ldots, \widetilde{Z}_{m,c}^{(i)}]\right)_t$. The copula-consistency error is

$$\text{CCErr} \triangleq \frac{1}{mk} \sum_{t=1}^m \sum_{c=1}^k \text{Var}_{i \in [d]}\left(U_{t,c}^{(i)}\right). \tag{91}$$

A smaller value indicates stronger cross-modal agreement in projected copula ranks.

**Aggregation.** ProcRes and CCErr are computed per batch. For offline evaluation, they are averaged over test batches and then over seeds. For streaming evaluation, the raw per-batch values and fixed-decay EMA values are recorded using the same stream order as the main gate diagnostics.

### G.6. Complete Mini-Batch Protocol

Algorithm 1 gives the SSCA mini-batch protocol. Hard coupling is the default setting used for efficiency. Soft coupling is used in ablations and in settings explicitly marked as soft-coupling runs.

---

**Algorithm 1** Stable Spectral Copula Alignment (SSCA) Mini-Batch Protocol

---

**Require:** Mini-batch feature matrices $\{H^{(i)}\}_{i=1}^d$; shared dimension $k = 128$; $S = 64$; $S_\tau = 16$; $\alpha_{\text{clip}} = 0.005$; $\lambda_{\text{stab}} = 0.01$; $\tau_{\text{dep}} = 0.1$; $\tau_{\text{rank}} = 0.1$; $T_{\text{out}} = 2$; coupling mode in $\{\text{HARD}, \text{SOFT}\}$; rank mode in $\{\text{SOFTRANK}, \text{SKETCH}\}$.

**Ensure:** Updated projections $\{W^{(i)}\}$ and batch diagnostics.

 1: **for** $t = 1$ to $T_{\text{out}}$ **do**
 2:     **Marginal stabilization:** compute $G^{(i)} \leftarrow \mathcal{T}^{(i)}(H^{(i)})$ using soft-rank during training or sketch mode during deployment.
 3:     **Projection:** compute $Z_0^{(i)} \leftarrow G^{(i)} W^{(i)}$ for all modalities.
 4:     **Dependence weights:** estimate $\widehat{\tau}_{ij}$ by Kendall's $\tau$ on $S_\tau$ one-dimensional projections; set $\omega_{ij} \propto \exp([\widehat{\tau}_{ij}]_+/\tau_{\text{dep}})$ and $w_i \propto \sum_{j \neq i}[\widehat{\tau}_{ij}]_+$.
 5:     **Hub quantiles:** sample $S$ directions $\{\theta_s\}$; compute $\widehat{Q}_{\theta_s}^{(i)}$; set $Q_{\theta_s}^{(B)}(u) = \sum_i w_i \widehat{Q}_{\theta_s}^{(i)}(u)$.
 6:     **Hub coupling:** compute $\beta_i$ using Eq. (74); choose $i^\star$ using Eq. (75); build $\widetilde{P}_i$ using Eq. (76); obtain $P_i$ by Eq. (77) or $\Pi_i$ by Eq. (78).
 7:     **Coupled features:** compute $\widetilde{G}^{(i)} \leftarrow P_i G^{(i)}$ for soft coupling or $\widetilde{G}^{(i)} \leftarrow \Pi_i G^{(i)}$ for hard coupling.
 8:     **Centering:** compute $\bar{G}^{(i)} \leftarrow \widetilde{G}^{(i)} - \frac{1}{m}\mathbf{1}\mathbf{1}^\top \widetilde{G}^{(i)}$.
 9:     **Spectral operator:** compute $C_{ij} \leftarrow m^{-1}\bar{G}^{(i)\top}\bar{G}^{(j)}$ and $\Sigma_i \leftarrow m^{-1}\bar{G}^{(i)\top}\bar{G}^{(i)} + \lambda_{\text{stab}}I$; set $R_i = \Sigma_i^{-1/2}$ and define $\mathcal{M}_{ij} = \omega_{ij} R_i C_{ij} R_j$ for $i \neq j$, with $\mathcal{M}_{ii} = 0$.
10:     **Eigenspace update:** compute the top-$k$ eigenspace $\widehat{U}$ of $\mathcal{M}$ by an implicit eigensolver; set each $W^{(i)} \leftarrow \text{qf}(R_i\widehat{U}_i)$, where $\widehat{U}_i$ is the block associated with modality $i$.
11:     **Diagnostics:** record Eqs. (80)–(88), ProcRes in Eq. (90), and CCErr in Eq. (91).
12: **end for**

---

## H. Computational Complexity and Profiling

### H.1. Asymptotic Cost per Mini-Batch

Let $m$ be the mini-batch size, $d$ the number of modalities, $k = 128$ the shared embedding dimension, $S = 64$ the number of sliced-Wasserstein directions, $S_\tau = 16$ the number of directions used for dependence estimation, and $n_{\text{iter}}$ the number of eigensolver iterations. Let $q_i$ denote the feature dimension entering the rank-stabilization step for modality $i$, and define $Q = \sum_{i=1}^d q_i$. When stabilization is applied before projection, $q_i = p_i$; when an ablation applies stabilization after projection, $q_i = k$.

**Marginal stabilization.** Coordinate-wise exact sorting costs $\mathcal{O}(Q\, m \log m)$. Differentiable soft-sorting kernels may form pairwise intermediate tensors, giving a practical upper cost $\mathcal{O}(Q\, m^2)$, with memory proportional to the number of retained pairwise buffers.

**Dependence estimation.** Kendall-type dependence weights are estimated from $S_\tau$ one-dimensional projections for each modality pair. The sorting-dominated cost is $\mathcal{O}(d^2 S_\tau m \log m)$.

**Hub coupling.** For each slicing direction, the projected samples are sorted for all modalities, yielding $\mathcal{O}(dSm \log m)$ for the sorting stage. If directional couplings are accumulated as sparse index maps, averaging costs $\mathcal{O}(dSm)$. If dense coupling matrices are materialized, the storage and accumulation cost becomes $\mathcal{O}(dSm^2)$. Soft coupling adds Sinkhorn normalization, $\mathcal{O}(d\, n_{\text{sk}}\, m^2)$, where $n_{\text{sk}}$ is the number of Sinkhorn iterations. Hard coupling rounds the averaged plan to a permutation; exact Hungarian rounding costs $\mathcal{O}(dm^3)$, while the default batched implementation uses the same averaged

plan and is profiled directly in Table 19.

**Spectral step.** The block-spectral step uses implicit matrix-vector products. When the spectral operator is applied through the shared $k$-dimensional projected blocks, the dominant eigensolver cost is $\mathcal{O}\left(n_{\text{iter}}\, d^2\, m\, k\right)$. For raw-feature blocks, $k$ is replaced by the working feature dimension. The implicit form avoids explicitly constructing a full dense block covariance matrix across all raw feature dimensions.

### H.2. Profiling Methodology

The profiling in Table 19 measures the alignment stage under the frozen-feature protocol. Encoder feature extraction and data loading are excluded. Each timing includes the forward pass, backward pass, and optimizer step for the alignment objective. The reported memory is peak allocated GPU memory during the profiled alignment stage.

**Memory accounting.** Soft-rank mode can allocate GB-scale memory because practical soft-sorting routines may retain pairwise difference tensors, stabilized exponentials, and backward buffers. Mixed-precision execution reduces some tensors to FP16, while selected accumulators remain FP32 for numerical stability. The reported allocated peaks therefore reflect the actual tensor footprint of the alignment stage rather than encoder or input-pipeline memory.

### H.3. Throughput Definition

Throughput is computed as

$$\text{Throughput} = \frac{\#\text{examples}}{\text{time}(\text{forward} + \text{backward} + \text{optimizer step})}. \tag{92}$$

Measurements are averaged over 1000 iterations after 200 warm-up iterations. The batch size is fixed to $m = 256$ unless a dataset-specific constraint requires a smaller batch. This definition matches the scope of the main profiling claim: computational cost is assessed for the alignment stage and excludes end-to-end foundation-model pretraining.

## I. Experimental Protocols and Reproducibility

### I.1. Datasets, Splits, Tasks, and Frozen Encoders

All benchmarks use fixed train, validation, and test splits. Task definitions, label mappings, class counts, and evaluation metrics are fixed before training and shared by all methods. This ensures that reported accuracies and retrieval metrics are computed on the same label or query space across SSCA and baselines.

**Multimodal classification.** For MOSEI, MELD, and MOSI, the text encoder uses a `bert-base-uncased` CLS feature of dimension 768, followed by $L_2$ normalization. Audio uses VGGish features of dimension 128, followed by $L_2$ normalization. Video uses FACET/OpenFace-style features of dimension 47, followed by $L_2$ normalization.

**Image–text retrieval.** For MSCOCO, Flickr30k, and CC3M-500K retrieval settings, frozen image and text features are extracted once and then used by all alignment methods under the same frozen-feature protocol. The projection heads map frozen features to the shared dimension $k = 128$.

### I.2. Training Schedule

All methods are trained with frozen encoders and trainable projection heads $W^{(i)}$. Unless a baseline definition requires a different form, the projection head is linear and all methods share the same optimizer family, validation schedule, and stopping rule.

The default schedule is:

optimizer: AdamW;

weight decay: $1 \times 10^{-4}$;

learning rate: fixed per dataset according to Appendix J.10;

early stopping: best validation metric matching the reported test metric;

batch size: $m = 256$, unless memory or dataset constraints require a smaller batch.

This policy matches model capacity and update budget at the projection-alignment level.

## I.3. Input Preprocessing Constants

Table 10 lists the fixed preprocessing constants used before frozen feature extraction.

*Table 10.* Input preprocessing constants used before frozen feature extraction.

| Modality | Constant | Value |
|----------|----------|-------|
| Audio | Sample rate | 16 kHz |
| Audio | Normalization range | $[-1, 1]$ |
| Video | Frame sampling | Uniform |
| Text | Tokenizer | BERT WordPiece |
| Text | Max length | 50 tokens, tail truncation |
| Image | Resize | $224 \times 224$ |

## I.4. Monotone Distortion Suite

The primary in-scope distortion is affine scaling:

$$\phi_{\text{scale}}(x) = 10x. \tag{93}$$

When monotone-sweep curves are used, the scaling factor is varied over a fixed grid while the insertion point is kept identical across all methods.

Additional monotone feature distortions are

$$\phi_{\log}(x) = \text{sign}(x) \odot \log(1 + |x|), \tag{94}$$
$$\phi_{\tanh}(x) = \tanh(\beta x), \qquad \beta = 2.0. \tag{95}$$

All distortions are applied at the task-level representation $h^{(i)}$ defined in Appendix G.2. The insertion point is identical for SSCA and all baselines.

## I.5. Missing-Modality Protocol

Let $\mathbf{a} \in \{0, 1\}^d$ denote the modality-availability mask. Each modality is independently observed with probability $1 - p_{\text{miss}}$:

$$a_i \sim \text{Bernoulli}(1 - p_{\text{miss}}), \qquad p_{\text{miss}} = 0.7, \tag{96}$$

and the mask is resampled until at least one modality is available:

$$\sum_{i=1}^{d} a_i \geq 1. \tag{97}$$

Available-only fusion is

$$\bar{h}_{\text{AO}} = \frac{1}{\sum_{i=1}^{d} a_i} \sum_{i=1}^{d} a_i h^{(i)}. \tag{98}$$

## I.6. Mask-Aware Standardized Imputation

Observed embeddings are zero-filled according to the availability mask:

$$\widetilde{h}^{(i)} = a_i h^{(i)}. \tag{99}$$

For target modality $j$, the imputer input is

$$u_j = [\widetilde{h}^{(1)}; \ldots; \widetilde{h}^{(j-1)}; \widetilde{h}^{(j+1)}; \ldots; \widetilde{h}^{(d)}; \mathbf{a}] \in \mathbb{R}^{(d-1)k+d}. \tag{100}$$

The ridge imputer is trained on the training split only:

$$R_j = \arg\min_R \sum_{n \in \mathcal{D}_{\text{train}}} \left\| h_n^{(j)} - R u_{j,n} \right\|_2^2 + \lambda_{\text{SI}} \|R\|_F^2, \qquad \lambda_{\text{SI}} = 0.1. \tag{101}$$

The standardized-imputation fusion vector is

$$\bar{h}_{\text{SI}} = \frac{1}{d} \sum_{i=1}^{d} h_{\text{SI}}^{(i)}, \tag{102}$$

where

$$h_{\text{SI}}^{(i)} = \begin{cases} h^{(i)}, & a_i = 1, \\ R_i u_i, & a_i = 0. \end{cases} \tag{103}$$

No validation or test labels are used to train the imputation maps.

## J. Streaming Sensitivity Protocols

This section defines comparable streaming protocols for short-horizon and long-stream evaluation. The goal is to separate structural drift behavior from artifacts caused by stream length, burn-in, or repeated passes.

### J.1. Deterministic Cycling and Pass Accounting

A deterministic stream is built by iterating through the test split in a fixed order and forming batches of size $m$. Let $N_{\text{test}}$ be the test-set size. The number of batches in one pass is

$$B_{\text{pass}} \triangleq \left\lceil \frac{N_{\text{test}}}{m} \right\rceil. \tag{104}$$

For a stream of length $T$, the induced number of passes is

$$\#\text{passes}(T) \triangleq \frac{T}{B_{\text{pass}}}. \tag{105}$$

### J.2. Matched Drift Placement by Normalized Time

Drift indices are defined by normalized stream time:

$$t_1(T) \triangleq \lfloor 0.33T \rfloor, \qquad t_2(T) \triangleq \lfloor 0.66T \rfloor. \tag{106}$$

For $T = 6000$, this gives $(t_1, t_2) = (2000, 4000)$. Table 11 summarizes the two protocol families.

*Table 11.* Streaming sensitivity protocol families. Drift indices follow Eq. (106), which aligns drift placement by normalized stream time.

| Protocol family | Stream length $T$ | Drift indices | Primary role |
|---|---|---|---|
| Short-horizon | $T \in \{1, 3, 5\} \cdot B_{\text{pass}}$ | $t_1(T), t_2(T)$ | Tests behavior under short deployment windows and few repeated passes. |
| Long-stream | $T = 6000$ | $(2000, 4000)$ | Tests cumulative stability under long deployment streams. |

### J.3. Prequential Evaluation

At each streaming step $t$, the model first predicts on the current batch and records the metric. It then applies label-free alignment updates using the current batch inputs only. Updates use no labels, and the classifier is kept fixed. This is a causal test-then-update protocol.

## J.4. Label-Free Online Update Rule

For each incoming batch, the alignment parameters $\theta$ receive $G$ gradient steps:

$$\theta \leftarrow \theta - \eta \nabla_\theta \mathcal{L}_{\text{align}}(\{h^{(i)}\}; \theta), \qquad \eta = 10^{-4}. \tag{107}$$

The default update budget is $G = 10$. The no-update control uses $G = 0$, with the classifier frozen in both settings.

## J.5. Reporting Items

Each streaming protocol records the following quantities:

raw accuracy trajectory $\{a_t\}_{t=1}^T$;

EMA accuracy trajectory with decay $0.1$;

recovery time after each drift, defined in Appendix J.7;

post-drift worst-case value within a fixed window $\Delta$, specified by the experiment protocol;

gate decisions, fallback events, and diagnostic proxies.

## J.6. Streaming Sanity Checks

Two controls are used to detect stream-order artifacts. The order-randomized control deterministically shuffles the stream once with a fixed seed while preserving batch size. The single-pass control sets $T = B_{\text{pass}}$, so each test example appears once. These controls check whether the observed behavior depends on repeated cycling through the same stream.

## J.7. Recovery Time Metric

Let $a_t$ be the batch accuracy and let $\text{EMA}(a_t)$ denote the EMA with decay $0.1$. For a drift point $t_0$, define the pre-drift EMA reference as $\text{EMA}(a_{t_0^-})$. The recovery time is

$$\tau_{\text{rec}} = \min \left\{ t \geq t_0 : \text{EMA}(a_t) \geq 0.95 \cdot \text{EMA}(a_{t_0^-}) \right\}. \tag{108}$$

If the evaluated stream ends before the threshold is reached, $\tau_{\text{rec}}$ is recorded as censored at the stream endpoint.

## J.8. Logging Contract for Closed-Loop Auditability

The streaming protocol records the same diagnostic quantities used by the gate. For each batch $t$, the recorded fields are:

raw metric $a_t$ and EMA metric $\text{EMA}(a_t)$;

drift indicator and drift index, when applicable;

raw proxies $\widehat{\varepsilon}_{\text{rank}}(t)$, $\widehat{\varepsilon}_{\text{qnt}}(t)$ when enabled, $\widehat{\varepsilon}_{\text{ds}}(t)$ for soft coupling, $\widehat{\varepsilon}_{\text{rnd}}(t)$, $\widehat{\varepsilon}_{\text{hub}}(t)$, $\widehat{\varepsilon}_{\text{samp}}(t)$, and $\widehat{\varepsilon}_{\text{num}}(t)$;

empirical eigengap $\widehat{\gamma}(t)$;

eigengap-normalized proxies $\widehat{\varepsilon}_\bullet(t)/(\widehat{\gamma}(t) + 10^{-8})$ for the proxy terms used by the operational bound.

These fields support the closed-loop interpretation from observed degradation to quantitative localization and gate decision.

**Seed-level aggregation.** Curves are first computed per seed and then averaged across five seeds. Uncertainty bands report $\pm 1$ standard deviation. When EMA curves are plotted, EMA smoothing is applied per seed before cross-seed averaging.

## J.9. Statistical Testing Details

Paired bootstrap uses 1000 resamples and Holm–Bonferroni correction within each metric family.

**Classification.** Examples are resampled with replacement. Accuracy is computed for each resample, and paired differences are compared across methods.

**Retrieval.** Queries are resampled with replacement. Image-to-text retrieval resamples image queries, and text-to-image retrieval resamples text queries. Recall@K is computed for each bootstrap resample.

## J.10. Hyperparameter Summary

The following SSCA hyperparameters are fixed across experiments unless explicitly overridden:

$$k = 128, \quad S = 64, \quad S_\tau = 16, \quad \alpha_{\mathrm{clip}} = 0.005, \quad \lambda_{\mathrm{stab}} = 0.01, \quad \tau_{\mathrm{dep}} = 0.1, \quad \tau_{\mathrm{rank}} = 0.1, \quad T_{\mathrm{out}} = 2.$$

The sketch size is $s_{\mathrm{sketch}} = 400$. The sketch update step $\eta_{\mathrm{sketch}}$ is fixed for a given experiment. The imputation regularization is $\lambda_{\mathrm{SI}} = 0.1$.

# K. Operational Contract: Practical Thresholds and Failure Boundary

This appendix specifies the operational contract used by SSCA. The contract maps observable diagnostics to a stability gate, defines label-free threshold calibration, and states the boundary cases where rank-based invariance should not be trusted. The formal contract applies to approximately coordinate-wise, orientation-preserving monotone marginal distortions in feature space. Non-monotone transformations, severe ties, weak eigengaps, and unstable coupling regimes are handled by diagnostic rejection and no-update fallback.

## K.1. A Thresholded Stability Gate from Observable Proxies

Let $\widehat{\varepsilon}_{\mathrm{rank}}, \widehat{\varepsilon}_{\mathrm{cpl}}, \widehat{\varepsilon}_{\mathrm{samp}}, \widehat{\varepsilon}_{\mathrm{num}}$ be the empirical proxy terms used in the main diagnostic dictionary. Let $\widehat{\varepsilon}_{\mathrm{qnt}}$ denote the quantile-sketch proxy in Appendix G.4; it is used as a rank-side contribution and is zero in pure soft-rank mode. Define

$$\widehat{\varepsilon}^{+}_{\mathrm{rank}} \triangleq \widehat{\varepsilon}_{\mathrm{rank}} + \widehat{\varepsilon}_{\mathrm{qnt}}. \tag{109}$$

For calibrated coefficients $c_{\mathrm{rank}}, c_{\mathrm{cpl}}, c_{\mathrm{samp}}, c_{\mathrm{num}}$ and empirical eigengap $\widehat{\gamma}$, define the operational proxy risk

$$\widehat{\Delta} \triangleq \frac{c_{\mathrm{rank}}\widehat{\varepsilon}^{+}_{\mathrm{rank}} + c_{\mathrm{cpl}}\widehat{\varepsilon}_{\mathrm{cpl}} + c_{\mathrm{samp}}\widehat{\varepsilon}_{\mathrm{samp}} + c_{\mathrm{num}}\widehat{\varepsilon}_{\mathrm{num}}}{\widehat{\gamma} + \epsilon_\gamma}, \qquad \epsilon_\gamma = 10^{-8}. \tag{110}$$

The auditing gate is

$$\mathsf{Gate}(\tau, \gamma_{\min}) = \mathbf{1}\!\left[\widehat{\Delta} \leq \tau \ \wedge \ \widehat{\gamma} \geq \gamma_{\min}\right]. \tag{111}$$

**Proposition K.1** (Thresholded auditing gate). *Fix a tolerance $\tau > 0$ and eigengap floor $\gamma_{\min} > 0$. Assume that the orientation-preserving monotone-drift conditions of Theorem 3.1 hold, the calibrated proxy envelope is valid on the healthy calibration regime, and the current batch satisfies $\mathsf{Gate}(\tau, \gamma_{\min}) = 1$. Then the operational proxy risk in Eq. (110) supplies a sufficient certificate for bounded subspace perturbation at the calibrated tolerance level.*

**No-update fallback.** When $\mathsf{Gate} = 0$, SSCA suppresses the alignment update for the current batch. The protocol records the dominant diagnostics and uses the compute-matched no-update action for that batch. This fallback gives a conservative control action outside the contract. It only prevents the method from forcing an update when the calibrated proxy risk or the eigengap indicates an uncertified regime.

**Label-free threshold calibration.** The pair $(\tau, \gamma_{\min})$ is set from a source validation segment or a pre-drift unlabeled stream. The calibration procedure estimates the healthy distribution of $\widehat{\Delta}$ and $\widehat{\gamma}$, sets $\tau$ to a high quantile of $\widehat{\Delta}$, and sets $\gamma_{\min}$ to a conservative lower quantile of $\widehat{\gamma}$. No target labels are used. The calibrated gate therefore rejects batches whose proxy risk or eigengap deviates materially from the healthy operating regime.

## K.2. Gate False-Positive/False-Negative Curves under Label-Free Calibration

This subsection reports error-rate curves for the auditing gate under label-free quantile calibration. The calibration level $\delta$ is swept while all other components remain fixed. The false-positive rate is the rate of accepting out-of-scope drifted batches, and the false-negative rate is the rate of rejecting in-scope monotone-drift batches. Each curve reports the maximum error rate across five drift-strength levels. Figure 3 shows the resulting calibration curves. Figure 4 gives an out-of-scope boundary case. The non-monotone distortion increases order violations, the gate rejects unstable updates, and the fallback policy avoids the error growth caused by always accepting updates. Table 12 compares the contract gate against common monitoring rules on the same mixed-drift protocol.

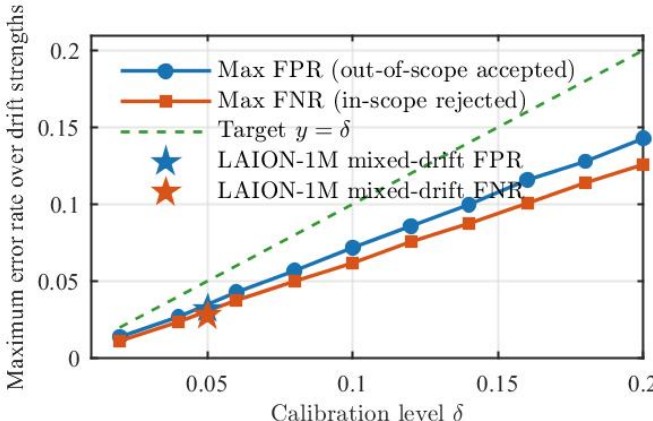

*Figure 3.* Auditing-gate calibration curves: maximum FPR/FNR across drift-strength levels versus calibration level $\delta$ in the synthetic stress test, with the target line $y = \delta$. The marker at $\delta = 0.05$ overlays the LAION-1M mixed-drift empirical point reported in Table 12: FPR = 3.2% and FNR = 2.8%.

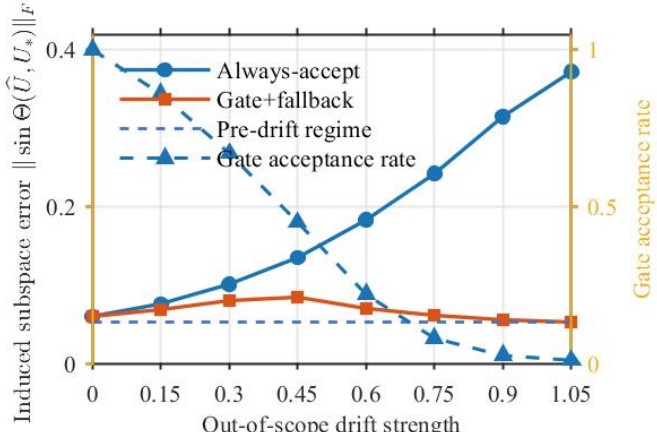

*Figure 4.* **Out-of-scope synthetic counterexample with gate-triggered fallback.** A deliberately non-monotone folding/saturation distortion violates the monotone marginal contract and induces high order violations. As drift strengthens, the gate rejects updates and the protocol uses the no-update fallback. The plot reports gate acceptance rate and induced subspace error under always-accept versus gate+fallback, showing that the gate prevents error escalation caused by unsupported updates.

*Table 12.* Contract validity verification on LAION-1M mixed-scope drifts. Lower is better for FPR/FNR; higher is better for intervention accuracy.

| Detector | FPR $\downarrow$ | FNR $\downarrow$ | Intervention Acc. $\uparrow$ |
|---|---|---|---|
| SSCA contract gate | 3.2% | 2.8% | 94.6% |
| w/o calibrated contract | 8.7% | 9.2% | 86.0% |
| MMD shift detector | 12.5% | 6.7% | 88.0% |
| Logit entropy monitor | 14.1% | 7.9% | 85.4% |
| Cosine-similarity threshold | 9.8% | 5.2% | 90.1% |

### K.3. Gate Coverage Metrics on Real-World Drifts

This subsection defines the statistics used to report gate coverage on the real-world drift suite. The corresponding quantitative results appear in Table 6.

**Per-episode quantities.** Consider a drift episode $t$, such as a streaming batch or a fixed corruption setting. Let $g_t \in \{0, 1\}$ denote the gate decision, where $g_t = 1$ applies Stability Mode and $g_t = 0$ applies the no-update fallback. Let $\Delta_t^{\text{SSCA}}$ be the degradation under always-updating SSCA, $\Delta_t^{\text{FB}}$ the degradation under always-fallback, and $\Delta_t^{\text{Gated}}$ the degradation under the gated policy. All degradations use the main-paper definition: clean performance minus drifted performance under the

same evaluation protocol.

**Coverage, mis-trigger costs, and benefit.** The reported statistics are

$$\text{REFUSE}(\%) := 100 \cdot \frac{1}{T} \sum_{t=1}^{T} (1 - g_t), \tag{112}$$

$$\text{UNSAFEACCEPT}(\%) := 100 \cdot \frac{\sum_{t=1}^{T} \mathbb{K}\left[g_t = 1 \wedge \Delta_t^{\text{SSCA}} > \Delta_t^{\text{FB}} + \eta_{\text{unsafe}}\right]}{\sum_{t=1}^{T} \mathbb{K}[g_t = 1] + 10^{-12}}, \tag{113}$$

$$\text{REGRET} := \frac{\sum_{t=1}^{T} \mathbb{K}[g_t = 0] \left(\Delta_t^{\text{FB}} - \Delta_t^{\text{SSCA}}\right)_+}{\sum_{t=1}^{T} \mathbb{K}[g_t = 0] + 10^{-12}}, \tag{114}$$

$$\text{BENEFIT} := \frac{1}{T} \sum_{t=1}^{T} \left(\Delta_t^{\text{SSCA}} - \Delta_t^{\text{Gated}}\right). \tag{115}$$

Here $(a)_+ = \max\{a, 0\}$, and $\eta_{\text{unsafe}} = 0.5$ metric points prevents negligible differences from being counted as unsafe acceptances.

**Aggregation protocol.** For each task, the statistics are aggregated over drift types and strengths when applicable. The values $\Delta_t^{\text{SSCA}}$, $\Delta_t^{\text{FB}}$, and $\Delta_t^{\text{Gated}}$ are measured under the same drift realization and seed. The fallback baseline is compute matched by using the same frozen-feature protocol, batch size, and update budget accounting.

**Interpretation.** REFUSE measures how often the policy leaves Stability Mode. UNSAFEACCEPT measures the empirical risk of accepting when fallback would have been safer. REGRET measures the cost of unnecessary refusal. BENEFIT measures the net degradation reduction of the gated policy relative to always-updating SSCA. A useful contract has low UNSAFEACCEPT, small REGRET, positive BENEFIT, and a moderate refusal rate.

### K.4. Boundary Conditions: Non-Monotone Distortions and Heavy Ties

The formal contract targets orientation-preserving monotone marginal distortions. Two boundary cases are important in practice: non-monotone transforms and heavy ties.

**Proposition K.2** (Failure boundary under non-monotone marginal transformations). *If a marginal transformation $g$ is non-monotone on a set of non-negligible probability mass, rank order can change. There exist random variables $X$ for which $\rho_{\text{Spearman}}(X, g(X)) < 1$, and the transformation $(X, Y) \mapsto (g(X), Y)$ can change the copula of $(X, Y)$. Any method whose invariance relies on monotone rank preservation can therefore degrade in this regime.*

**Proposition K.3** (Heavy-tie boundary under quantization or clipping). *Let $S$ be a scalar score obtained from a continuous latent variable by quantization or hard clipping. Let $S'$ be an independent copy and define the tie probability $p_{\text{tie}} \triangleq \Pr(S = S')$. When $p_{\text{tie}}$ is large, multiple total orders are compatible with the observed scores. Rank-based estimators then incur irreducible ambiguity proportional to the tie mass, which appears in the tie-aware rank diagnostics and can cause the auditing gate in Eq.* (111) *to reject operation.*

### K.5. Diagnostics-to-Remediation: A Protocol-Level Action Map

When the gate fails, the diagnostic dictionary identifies which component dominates the proxy risk. Algorithm 2 defines a compute-bounded remediation map. If the adjusted configuration still fails the gate, the protocol uses the no-update fallback.

## L. Ablations, Sensitivity, and Boundary Conditions

All ablations use the frozen-feature protocol and the same projection-level capacity unless otherwise stated. Table 13 summarizes the qualitative role of each component and points to the quantitative tables.

**Multiway coherence metrics.** The protocol reports ProcRes and CCErr as multiway consistency diagnostics. Their definitions and aggregation rules are given in Appendix G.5. Both metrics are computed at the batch level and used to relate cycle coherence to downstream robustness.

---

**Algorithm 2** Diagnostics-Driven Remediation

---

**Require:** Unlabeled mini-batch feature matrices $\{H^{(i)}\}_{i=1}^{d}$; current SSCA state; calibrated thresholds.
 1: Compute $\widehat{\varepsilon}_{\mathrm{rank}}, \widehat{\varepsilon}_{\mathrm{qnt}}, \widehat{\varepsilon}_{\mathrm{cpl}}, \widehat{\varepsilon}_{\mathrm{samp}}, \widehat{\varepsilon}_{\mathrm{num}}$, and $\widehat{\gamma}$.
 2: **if** $\widehat{\varepsilon}_{\mathrm{rank}}$ exceeds its threshold **then**
 3:     Increase $\tau_{\mathrm{rank}}$ within the calibrated range and increase $\alpha_{\mathrm{clip}}$ to reduce tail amplification.
 4: **end if**
 5: **if** $\widehat{\varepsilon}_{\mathrm{qnt}}$ exceeds its threshold **then**
 6:     Increase sketch size or update frequency for rank/quantile estimation.
 7: **end if**
 8: **if** $\widehat{\varepsilon}_{\mathrm{cpl}}$ exceeds its threshold **then**
 9:     Increase the number of slicing directions $S$ or strengthen hub regularization.
10: **end if**
11: **if** $\widehat{\varepsilon}_{\mathrm{samp}}$ exceeds its threshold **then**
12:     Increase batch size when available or use EMA averaging of cross-covariances.
13: **end if**
14: **if** $\widehat{\varepsilon}_{\mathrm{num}}$ exceeds its threshold **then**
15:     Increase $\lambda_{\mathrm{stab}}$ or tighten the eigensolver tolerance.
16: **end if**
17: Recompute the gate under the adjusted configuration.
18: **if** $\mathsf{Gate}(\tau, \gamma_{\min}) = 1$ **then**
19:     Apply the standard SSCA update under the same online-update budget.
20: **else**
21:     Use the no-update fallback for this batch.
22: **end if**

---

*Table 13.* Core ablations and their observed effects.

| Ablation | Observed effect |
|---|---|
| Remove soft-rank Gaussianization | Under monotone distortions, degradation increases relative to the full method (Table 7). |
| Hub coupling $\rightarrow$ pairwise alignment | Multiway coherence deteriorates and robustness under perturbation/drift weakens (Tables 9 and 7). |
| $\lambda_{\mathrm{stab}} = 0$ | Perturbed performance decreases and numerical residuals increase, weakening the stability certificate (Table 7). |
| Uniform weights instead of dependence weighting | Robustness drops under shifts where noisy or spurious dependence affects the alignment objective (Table 7). |

### L.1. Failure Modes and Boundary Conditions

SSCA targets orientation-preserving monotone marginal distortions. Performance can degrade outside this scope.

**Non-monotone transformations.** Large geometric changes, adversarial token permutations, foldback saturation, or other local order reversals can break rank preservation. In this regime, the gate is expected to reject updates when the diagnostics indicate an unstable rank or coupling regime.

**Unstable CDF regimes.** Heavy-tailed noise, severe quantization, or very small batches can make empirical CDF estimates unstable. The corresponding boundary is reflected in elevated $\widehat{\varepsilon}_{\mathrm{rank}}$, $\widehat{\varepsilon}_{\mathrm{qnt}}$, or tie-mass diagnostics.

**Weak shared signal.** When cross-modal dependence is weak, Kendall-type dependence weights become noisy and the spectral eigengap can shrink. This regime is indicated by unstable dependence weights and a small $\widehat{\gamma}$.

**Directional coupling disagreement.** When one-dimensional OT permutations disagree strongly across slicing directions, the averaged plan $\widetilde{P}_i$ becomes diffuse and rounding becomes unstable. This regime is detected by elevated $\widehat{\varepsilon}_{\mathrm{rnd}}$ and a small hub margin $\widehat{\varepsilon}_{\mathrm{hub}}$.

# M. Additional Experimental Results

## M.1. Full Retrieval Results

Table 14 reports complete clean image-to-text retrieval results on MSCOCO and Flickr30k. All methods use the same frozen-feature protocol, trainable projection dimension, and evaluation split. Text-to-image metrics are computed by the same retrieval routine when enabled.

*Table 14.* Complete clean image-to-text retrieval results on MSCOCO and Flickr30k (mean±std over 5 random seeds). † indicates $p < 0.05$ versus the best non-SSCA baseline using paired bootstrap with Holm–Bonferroni correction.

| Method | MSCOCO (Image→Text) | | | Flickr30k (Image→Text) | | |
|---|---|---|---|---|---|---|
| | R@1 | R@5 | R@10 | R@1 | R@5 | R@10 |
| InfoNCE | $53.2 \pm 0.6$ | $79.8 \pm 0.5$ | $89.2 \pm 0.3$ | $68.1 \pm 0.7$ | $90.5 \pm 0.4$ | $95.8 \pm 0.2$ |
| SigLIP | $54.1 \pm 0.6$ | $80.5 \pm 0.5$ | $89.7 \pm 0.3$ | $68.9 \pm 0.7$ | $91.0 \pm 0.4$ | $96.1 \pm 0.2$ |
| 3-View Contrastive | $53.8 \pm 0.6$ | $80.2 \pm 0.5$ | $89.5 \pm 0.3$ | $68.5 \pm 0.7$ | $90.8 \pm 0.4$ | $96.0 \pm 0.2$ |
| DCCA | $52.0 \pm 0.6$ | $78.5 \pm 0.5$ | $88.2 \pm 0.3$ | $66.9 \pm 0.7$ | $89.4 \pm 0.4$ | $95.0 \pm 0.2$ |
| GCCA | $52.5 \pm 0.6$ | $78.9 \pm 0.5$ | $88.5 \pm 0.3$ | $67.2 \pm 0.7$ | $89.7 \pm 0.4$ | $95.2 \pm 0.2$ |
| SW-Align | $50.1 \pm 0.6$ | $77.5 \pm 0.5$ | $87.8 \pm 0.3$ | $65.8 \pm 0.7$ | $88.9 \pm 0.4$ | $94.7 \pm 0.2$ |
| Sinkhorn | $50.5 \pm 0.6$ | $77.8 \pm 0.5$ | $88.0 \pm 0.3$ | $66.1 \pm 0.7$ | $89.1 \pm 0.4$ | $94.9 \pm 0.2$ |
| SW-OT+CC | $50.7 \pm 0.6$ | $77.9 \pm 0.5$ | $88.1 \pm 0.3$ | $66.2 \pm 0.7$ | $89.2 \pm 0.4$ | $95.0 \pm 0.2$ |
| MMML | $54.5 \pm 0.5$ | $80.8 \pm 0.4$ | $89.9 \pm 0.3$ | $69.5 \pm 0.6$ | $91.5 \pm 0.3$ | $96.3 \pm 0.2$ |
| SSCA | $\mathbf{55.0 \pm 0.4}$† | $\mathbf{81.2 \pm 0.3}$† | $\mathbf{90.3 \pm 0.2}$† | $\mathbf{70.2 \pm 0.5}$† | $\mathbf{92.1 \pm 0.3}$† | $\mathbf{96.8 \pm 0.1}$† |
| SSCA+Contrastive | $\mathbf{55.3 \pm 0.4}$† | $\mathbf{81.5 \pm 0.3}$† | $\mathbf{90.5 \pm 0.2}$† | $\mathbf{70.5 \pm 0.5}$† | $\mathbf{92.3 \pm 0.3}$† | $\mathbf{96.9 \pm 0.1}$† |

## M.2. CLIP ViT-B/16 on CC3M-500K: Protocol and Full JPEG Sweep

**Protocol.** This experiment applies SSCA to frozen CLIP ViT-B/16 image–text embeddings and evaluates robustness on a 500K subset of CC3M (Sharma et al., 2018). The alignment stage uses the same optimization and regularization defaults as Sec. 5.1 under the frozen-feature protocol. JPEG compression is applied at test time with quality factors $Q \in \{70, 50, 30\}$. All values are mean±std over three random seeds. The degradation and reduction metrics follow Eqs. (30)–(31). Table 15 reports the full JPEG sweep.

*Table 15.* SSCA with frozen CLIP ViT-B/16 on CC3M-500K under JPEG compression. Reported values are Recall@1 (mean±std over three seeds). † indicates $p < 0.05$ versus CLIP+MMML.

| Method | Clean R@1 | JPEG Q=70 | JPEG Q=50 | JPEG Q=30 |
|---|---|---|---|---|
| CLIP Baseline | $58.7 \pm 0.5$ | $56.2 \pm 0.6$ | $53.6 \pm 0.6$ | $49.1 \pm 0.7$ |
| CLIP + SigLIP | $59.3 \pm 0.5$ | $57.0 \pm 0.5$ | $54.5 \pm 0.6$ | $50.2 \pm 0.7$ |
| CLIP + SW-Align | $58.9 \pm 0.5$ | $56.8 \pm 0.6$ | $54.2 \pm 0.6$ | $50.0 \pm 0.7$ |
| CLIP + MUG | $59.5 \pm 0.5$ | $57.3 \pm 0.5$ | $55.0 \pm 0.6$ | $51.0 \pm 0.6$ |
| CLIP + MMML | $59.8 \pm 0.4$ | $57.5 \pm 0.5$ | $55.9 \pm 0.5$ | $52.1 \pm 0.6$ |
| SSCA (CLIP) | $\mathbf{59.8 \pm 0.4}$ | $\mathbf{58.4 \pm 0.5}$ | $\mathbf{57.3 \pm 0.5}$† | $\mathbf{54.9 \pm 0.6}$† |

**Reduction at Q=50.** At JPEG Q=50, the CLIP baseline degradation is $58.7 - 53.6 = 5.1$, and the SSCA degradation is $59.8 - 57.3 = 2.5$. The degradation reduction is therefore $\mathrm{Red}(50) = 100 \times \frac{5.1 - 2.5}{5.1} \approx 51.0\%$.

## M.3. Monotone Distortion Suite on CMU-MOSEI and MELD

Table 16 reports additional non-affine monotone transformations beyond the main affine-scaling experiment.

## M.4. Cross-Dataset Transfer

Table 17 evaluates whether the learned alignment transfers across retrieval datasets without target adaptation.

*Table 16.* Performance under additional monotone distortions on CMU-MOSEI and MELD (mean±std over 5 seeds). Degradation $\Delta$ relative to clean accuracy is shown in parentheses.

| Method | Dataset | Clean (Acc%) | Log-Warping (Acc%) | Sigmoidal Saturation (Acc%) |
|---|---|---|---|---|
| InfoNCE | CMU-MOSEI | $78.9 \pm 0.5$ | $75.3 \pm 0.6$ (3.6) | $76.1 \pm 0.6$ (2.8) |
| | MELD | $59.8 \pm 0.6$ | $56.2 \pm 0.6$ (3.6) | $57.0 \pm 0.6$ (2.8) |
| SigLIP | CMU-MOSEI | $79.5 \pm 0.5$ | $76.0 \pm 0.6$ (3.5) | $76.8 \pm 0.6$ (2.7) |
| | MELD | $60.3 \pm 0.6$ | $56.8 \pm 0.6$ (3.5) | $57.6 \pm 0.6$ (2.7) |
| 3-View Contrastive | CMU-MOSEI | $79.8 \pm 0.5$ | $76.3 \pm 0.6$ (3.5) | $77.1 \pm 0.6$ (2.7) |
| | MELD | $60.5 \pm 0.6$ | $57.1 \pm 0.6$ (3.4) | $57.9 \pm 0.6$ (2.6) |
| GCCA | CMU-MOSEI | $77.6 \pm 0.6$ | $75.8 \pm 0.6$ (1.8) | $76.2 \pm 0.6$ (1.4) |
| | MELD | $59.0 \pm 0.7$ | $57.5 \pm 0.6$ (1.5) | $58.1 \pm 0.6$ (0.9) |
| SW-Align | CMU-MOSEI | $79.2 \pm 0.5$ | $77.0 \pm 0.6$ (2.2) | $77.8 \pm 0.6$ (1.4) |
| | MELD | $60.7 \pm 0.6$ | $58.3 \pm 0.6$ (2.4) | $59.0 \pm 0.6$ (1.7) |
| SW-OT+CC | CMU-MOSEI | $79.5 \pm 0.5$ | $77.3 \pm 0.6$ (2.2) | $78.1 \pm 0.6$ (1.4) |
| | MELD | $61.0 \pm 0.6$ | $58.6 \pm 0.6$ (2.4) | $59.3 \pm 0.6$ (1.7) |
| MMML | CMU-MOSEI | $86.7 \pm 0.3$ | $84.5 \pm 0.4$ (2.2) | $85.1 \pm 0.4$ (1.6) |
| | MELD | $66.7 \pm 0.4$ | $64.5 \pm 0.4$ (2.2) | $65.1 \pm 0.4$ (1.6) |
| SSCA | CMU-MOSEI | $\mathbf{87.8 \pm 0.3}$ | $\mathbf{86.2 \pm 0.3}$ **(1.6)** | $\mathbf{86.7 \pm 0.3}$ **(1.1)** |
| | MELD | $\mathbf{67.8 \pm 0.4}$ | $\mathbf{65.8 \pm 0.4}$ **(2.0)** | $\mathbf{66.5 \pm 0.4}$ **(1.3)** |

*Table 17.* Zero-shot transfer for image-text retrieval (mean±std over 5 seeds). Models are trained on the source dataset and evaluated on the target without adaptation. † indicates $p < 0.05$ versus the best non-SSCA baseline.

| Method | Recall@1 (%) | |
|---|---|---|
| | COCO $\rightarrow$ Flickr30k | Flickr30k $\rightarrow$ COCO |
| InfoNCE | $62.3 \pm 0.7$ | $48.1 \pm 0.7$ |
| SigLIP | $63.0 \pm 0.7$ | $48.9 \pm 0.7$ |
| 3-View Contrastive | $62.7 \pm 0.7$ | $48.5 \pm 0.7$ |
| DCCA | $61.2 \pm 0.7$ | $47.1 \pm 0.7$ |
| GCCA | $61.5 \pm 0.7$ | $47.3 \pm 0.7$ |
| SW-Align | $60.1 \pm 0.7$ | $46.2 \pm 0.7$ |
| Sinkhorn | $60.4 \pm 0.7$ | $46.5 \pm 0.7$ |
| SW-OT+CC | $60.5 \pm 0.7$ | $46.6 \pm 0.7$ |
| MMML | $63.8 \pm 0.6$ | $49.8 \pm 0.6$ |
| SSCA | $\mathbf{65.8 \pm 0.5}$† | $\mathbf{51.2 \pm 0.5}$† |
| SSCA+Contrastive | $\mathbf{66.1 \pm 0.5}$† | $\mathbf{51.5 \pm 0.5}$† |

## M.5. Retrieval Under Non-Affine Monotone Distortions

Table 18 reports image-to-text retrieval robustness under log-warping and sigmoidal saturation.

## N. Computational Efficiency Profiling

Table 19 reports alignment-stage profiling under the frozen-feature protocol. The measurements exclude data loading and encoder feature extraction. Throughput is measured in thousands of samples per second; wall-clock time is measured per mini-batch.

## O. Additional End-to-End and Control Experiments

### O.1. Critical Baseline Comparison

Table 20 reports the full results behind the critical baseline analysis. The comparison separates marginal stabilization from hub coupling and spectral learning.

*Table 18.* Image-text retrieval (R@1) under non-affine monotone distortions on MSCOCO and Flickr30k (mean±std over 5 seeds).

| Method | MSCOCO (R@1) | | Flickr30k (R@1) | |
|---|---|---|---|---|
| | Log-Warping | Sigmoidal Saturation | Log-Warping | Sigmoidal Saturation |
| InfoNCE | $51.8 \pm 0.6$ | $52.1 \pm 0.6$ | $66.5 \pm 0.7$ | $66.8 \pm 0.7$ |
| SigLIP | $52.7 \pm 0.6$ | $53.0 \pm 0.6$ | $67.3 \pm 0.7$ | $67.6 \pm 0.7$ |
| 3-View Contrastive | $52.5 \pm 0.6$ | $52.8 \pm 0.6$ | $67.0 \pm 0.7$ | $67.3 \pm 0.7$ |
| DCCA | $51.0 \pm 0.6$ | $51.3 \pm 0.6$ | $65.4 \pm 0.7$ | $65.7 \pm 0.7$ |
| GCCA | $51.3 \pm 0.6$ | $51.6 \pm 0.6$ | $65.7 \pm 0.7$ | $66.0 \pm 0.7$ |
| SW-Align | $49.8 \pm 0.6$ | $50.1 \pm 0.6$ | $64.3 \pm 0.7$ | $64.6 \pm 0.7$ |
| Sinkhorn | $50.1 \pm 0.6$ | $50.4 \pm 0.6$ | $64.6 \pm 0.7$ | $64.9 \pm 0.7$ |
| MMML | $53.2 \pm 0.5$ | $53.5 \pm 0.5$ | $68.1 \pm 0.6$ | $68.4 \pm 0.6$ |
| SSCA | $\mathbf{53.9 \pm 0.4}$ | $\mathbf{54.2 \pm 0.4}$ | $\mathbf{68.8 \pm 0.5}$ | $\mathbf{69.1 \pm 0.5}$ |

*Table 19.* Computational efficiency profiling for the alignment stage (batch size $m = 256$). Throughput is measured in thousands of samples per second. Memory is peak allocated GPU memory. Results are mean±std over 5 runs.

| Method | Throughput (k samples/s) | Memory (GB) | Wall-clock / batch (ms) |
|---|---|---|---|
| InfoNCE | $18.2 \pm 0.3$ | $5.1 \pm 0.1$ | $14.1 \pm 0.2$ |
| SigLIP | $18.5 \pm 0.3$ | $5.1 \pm 0.1$ | $13.8 \pm 0.2$ |
| 3-View Contrastive | $17.8 \pm 0.3$ | $5.3 \pm 0.1$ | $14.4 \pm 0.2$ |
| DCCA | $25.1 \pm 0.4$ | $3.9 \pm 0.1$ | $10.2 \pm 0.2$ |
| GCCA | $25.6 \pm 0.4$ | $3.8 \pm 0.1$ | $10.0 \pm 0.2$ |
| SW-Align | $20.1 \pm 0.3$ | $4.5 \pm 0.1$ | $12.7 \pm 0.2$ |
| Sinkhorn | $19.9 \pm 0.3$ | $4.6 \pm 0.1$ | $12.9 \pm 0.2$ |
| SW-OT+CC | $19.8 \pm 0.3$ | $4.6 \pm 0.1$ | $12.9 \pm 0.2$ |
| MMML | $19.2 \pm 0.3$ | $4.8 \pm 0.1$ | $13.3 \pm 0.2$ |
| SSCA (SoftRank) | $22.5 \pm 0.3$ | $4.2 \pm 0.1$ | $11.4 \pm 0.2$ |
| SSCA (Sketch) | $24.8 \pm 0.4$ | $3.9 \pm 0.1$ | $10.3 \pm 0.2$ |
| SSCA+Contrastive | $20.3 \pm 0.3$ | $4.6 \pm 0.1$ | $12.6 \pm 0.2$ |

## O.2. Raw-Pipeline Drift Suite on CMU-MOSEI and MELD

This suite applies preprocessing drift before feature extraction, including visual re-encoding, audio codec transcoding, and tokenizer/library changes. The same frozen encoders, projection heads, and adaptation budgets are used across methods. Table 21 reports representative results on CMU-MOSEI and MELD.

## O.3. Multi-Backbone Robustness across Task Families

This suite evaluates whether the same alignment protocol remains stable across encoder families and task types. Table 22 reports degradation under $10\times$ scaling for classification-style settings and JPEG Q=50 for retrieval or VQA settings.

## O.4. Compute-Matched Comparisons to Strong Controls

Table 23 compares SSCA with representative test-time adaptation, calibration, and distributionally robust learning controls under matched projection-level update budgets. All methods use the same number of gradient steps and the same trainable parameter set in the head or projection modules.

## O.5. Adapter Fine-Tuning Robustness

Table 24 reports an adapter control in which a lightweight LoRA adapter is trained on clean data and evaluated under $10\times$ scaling drift. The adapter rank is $r = 8$. The comparison uses the corresponding frozen-feature pipeline and reports clean accuracy and degradation.

*Table 20.* Critical baseline comparison on CMU-MOSEI and MELD under $10\times$ scaling. Degradation $\Delta$ is reported as mean±std. † indicates $p < 0.05$ versus the best critical baseline, Copula-only + SigLIP.

| Method | CMU-MOSEI $\Delta \downarrow$ | MELD $\Delta \downarrow$ |
|---|---|---|
| InfoNCE | $6.5 \pm 0.4$ | $4.8 \pm 0.3$ |
| SigLIP | $6.3 \pm 0.4$ | $4.7 \pm 0.3$ |
| Pairwise SW-OT | $3.5 \pm 0.2$ | $3.3 \pm 0.2$ |
| **Copula-only + InfoNCE** | $2.8 \pm 0.2$ | $2.7 \pm 0.2$ |
| **Copula-only + SigLIP** | $2.3 \pm 0.1$ | $2.2 \pm 0.1$ |
| **Pairwise SW-OT + Cycle Correction** | $2.4 \pm 0.1$ | $2.2 \pm 0.1$ |
| **SSCA (Full)** | $\mathbf{1.8 \pm 0.1}$† | $\mathbf{1.5 \pm 0.1}$† |

*Table 21.* Degradation $\Delta$ under raw-pipeline drifts applied before feature extraction on CMU-MOSEI and MELD. Lower is better.

| Method | CMU-MOSEI $\Delta \downarrow$ | | MELD $\Delta \downarrow$ | |
|---|---|---|---|---|
| | JPEG+Opus+Tokenizer | Avg. $\Delta$ | JPEG+Opus+Tokenizer | Avg. $\Delta$ |
| InfoNCE | $3.8 \pm 0.3$ | $3.9 \pm 0.2$ | $3.5 \pm 0.3$ | $3.6 \pm 0.2$ |
| SigLIP | $3.5 \pm 0.3$ | $3.6 \pm 0.2$ | $3.2 \pm 0.3$ | $3.3 \pm 0.2$ |
| MMML | $2.5 \pm 0.2$ | $2.6 \pm 0.1$ | $2.3 \pm 0.2$ | $2.4 \pm 0.1$ |
| TENT | $3.0 \pm 0.2$ | $3.1 \pm 0.1$ | $2.8 \pm 0.2$ | $2.9 \pm 0.1$ |
| CoTTA | $2.7 \pm 0.2$ | $2.8 \pm 0.1$ | $2.5 \pm 0.2$ | $2.6 \pm 0.1$ |
| VCCE | $2.9 \pm 0.2$ | $3.0 \pm 0.1$ | $2.7 \pm 0.2$ | $2.8 \pm 0.1$ |
| GroupDRO | $2.4 \pm 0.2$ | $2.5 \pm 0.1$ | $2.2 \pm 0.2$ | $2.3 \pm 0.1$ |
| **SSCA** | $\mathbf{1.9 \pm 0.1}$ | $\mathbf{2.0 \pm 0.1}$ | $\mathbf{1.7 \pm 0.1}$ | $\mathbf{1.8 \pm 0.1}$ |

*Table 22.* Multi-backbone, multi-task generalization. Degradation $\Delta$ is reported under the task-specific drift protocol.

| Backbone (Task) | Base Method | Base $\Delta$ | SSCA $\Delta$ |
|---|---|---|---|
| CLIP ViT-L/14 (Retrieval) | SigLIP | $4.5 \pm 0.3$ | $\mathbf{2.6 \pm 0.2}$ |
| ALIGN (Retrieval) | SigLIP | $4.8 \pm 0.3$ | $\mathbf{2.9 \pm 0.2}$ |
| BLIP (VQA) | MMML | $3.3 \pm 0.2$ | $\mathbf{2.0 \pm 0.1}$ |
| PANNs+CLIP (AV-Localization) | InfoNCE | $5.1 \pm 0.3$ | $\mathbf{3.2 \pm 0.2}$ |

*Table 23.* Compute-matched comparison with TTA/calibration methods on CMU-MOSEI and MELD. Degradation $\Delta$ is reported; lower is better.

| Method | CMU-MOSEI $\Delta \downarrow$ | | MELD $\Delta \downarrow$ | |
|---|---|---|---|---|
| | $10\times$ scaling | JPEG Q=50 | $10\times$ scaling | JPEG Q=50 |
| TENT | $2.6 \pm 0.2$ | $3.1 \pm 0.2$ | $2.6 \pm 0.2$ | $3.0 \pm 0.2$ |
| CoTTA | $2.4 \pm 0.2$ | $2.9 \pm 0.2$ | $2.4 \pm 0.2$ | $2.8 \pm 0.2$ |
| VCCE | $2.5 \pm 0.2$ | $3.0 \pm 0.2$ | $2.5 \pm 0.2$ | $2.9 \pm 0.2$ |
| MMML | $2.2 \pm 0.2$ | $2.7 \pm 0.2$ | $2.2 \pm 0.2$ | $2.7 \pm 0.2$ |
| GroupDRO | $2.0 \pm 0.2$ | $2.5 \pm 0.2$ | $1.9 \pm 0.2$ | $2.2 \pm 0.2$ |
| **SSCA** | $\mathbf{1.8 \pm 0.1}$ | $\mathbf{2.2 \pm 0.1}$ | $\mathbf{1.5 \pm 0.1}$ | $\mathbf{1.8 \pm 0.1}$ |

*Table 24.* LoRA($r = 8$) adapter control under $10\times$ scaling drift. Clean accuracy and degradation $\Delta$ are reported.

| Setting | MELD | | CMU-MOSEI | |
|---|---|---|---|---|
| | Clean $\uparrow$ | $\Delta \downarrow$ | Clean $\uparrow$ | $\Delta \downarrow$ |
| Frozen (no adapter) | $67.8 \pm 0.4$ | $1.6 \pm 0.2$ | $82.9 \pm 0.2$ | $1.7 \pm 0.1$ |
| LoRA($r = 8$) | $68.5 \pm 0.3$ | $1.4 \pm 0.2$ | $83.1 \pm 0.2$ | $1.6 \pm 0.1$ |

