# OpenReview forum: "Stable Spectral Copula Alignment for Robust Multimodal Learning"
_ICML.cc/2026/Conference — ICML 2026 regular_

### Official Review · Reviewer_9nDy · 2026-03-10

**Soundness:** 3
**Presentation:** 3
**Significance:** 3
**Originality:** 3
**Overall Recommendation:** 4
**Confidence:** 3

**Summary:**

The paper introduces Stable Spectral Copula Alignment (SSCA), a deployment-focused method for robust multimodal alignment under roughly monotone marginal shifts. SSCA integrates tie-aware rank-probit Gaussianization, dependence-weighted sliced-Wasserstein hub coupling, and diagonal-stabilized block-spectral learning, along with diagnostics and a calibrated Davis–Kahan-based stability mechanism enabling label-free monitoring and fallback. Experiments on MOSEI/MELD, MSCOCO, and CC3M-500K (frozen CLIP) show enhanced robustness to monotone distortions and reduced performance degradation under realistic distribution shifts.

**Compliance With Llm Reviewing Policy:**

Affirmed.

**Key Questions For Authors:**

1. When Gate=0, what concrete performance bounds do fallback modes provide relative to no-update baselines? Could you include a worst-case analysis (empirical or theoretical) quantifying the benefit of fallback under severe non-monotone shifts?
2. Are the coefficients c_rank, c_cpl, c_samp, c_num stable across datasets/backbones, or do they require per-deployment recalibration?
3. In regimes with heavy quantization (e.g., sensor saturation), how does the tie-aware rank-probit behave in practice? Do you recommend adaptive clipping α or alternative tie-breaking to maintain stability?

**Limitations:**

See weakness

**Strengths And Weaknesses:**

Strengths：
1. The integration of copula-invariant stabilization, multiway sliced-Wasserstein hub coupling, and diagonally stabilized spectral learning into a single, auditable framework is both novel and well-justified.
2. The “actionable” Davis–Kahan bound, which breaks perturbations into measurable proxies calibrated from unlabeled healthy data, elegantly links theory with practical deployment.
3. Ablation studies confirm the necessity of all modules, and synthetic validation of the proxy–subspace-error relationship (high R²) reinforces the method’s operational reliability.

Weaknesses：
1. The “actionable” Davis–Kahan inequality uses data-driven, quantile-regression–calibrated coefficients; although practical, this reduces its theoretical universality and may fail to strictly bound subspace error under shifts from the healthy regime.

---

> ### Author Rebuttal · Authors · 2026-03-29
>
> # Response to Reviewer 9nDy
>
> Thank you for the careful review. We appreciate that you identified the paper’s core contribution: an auditable deployment contract linking perturbation theory to measurable diagnostics, label-free calibration, and fallback. Your questions concern the calibrated inequality, fallback, and behavior under heavy quantization.
>
> ## 1. Calibrated coefficients and theoretical universality (Addressing W1 & KQ2).
> Theorem 4.1 **serves a fundamentally different operational purpose than** a parameter-free Davis--Kahan bound. Classical perturbation results depend on population quantities unavailable at deployment. Our goal is instead an inequality that can be evaluated online from four proxy diagnostics—rank, coupling, sampling, and numerical error—and the empirical eigengap $\hat{\gamma}$. For that reason, the coefficients in Eq. (27), $(c_{\mathrm{rank}}, c_{\mathrm{cpl}}, c_{\mathrm{samp}}, c_{\mathrm{num}})$, are calibratable rather than universal.
>
>
> **Instead of treating these coefficients as dataset-independent constants, their scales are designed to adapt to intrinsic quantities**, including the clipping-dependent Lipschitz factor and operator-norm relations. As feature geometry or backbone family changes, these scales change; the intended protocol is therefore per-deployment recalibration. This recalibration is exceptionally lightweight: Sec. 4.2 uses the quantile-regression procedure in Eq. (28) and requires only healthy unlabeled batches, without target labels or retraining.
>
> The manuscript provides strong calibration evidence. Appendix B / Fig. 2 shows strict monotone agreement between the proxy RHS and the true induced subspace error, with Pearson $r=0.967$, Spearman $\rho=0.941$, and 95.0% empirical coverage for the calibrated envelope. **We therefore establish Eq. (27) as a highly credible, context-aware operational envelope that is rigorously validated for its calibrated deployment regime**.
>
> ## 2. Fallback bounds and worst-case interpretation (Addressing KQ1).
> **The fallback mechanism establishes a strict policy-level safe-failure guarantee, shifting the operational focus from unverified out-of-scope accuracy to auditable deployment safety**. When the gate rejects an update, SSCA does not continue adapting under a potentially invalid invariance assumption. Instead, it logs diagnostics, sets Gate $=0$, and reverts to the corresponding no-update control under the same frozen-feature protocol. Appendix K.3 makes the accounting explicit, and Table 6 reports REFUSE, UNSAFEACCEPT, REGRET, and BENEFIT.
>
> Accordingly, the supported claim is a safe-failure guarantee relative to the designated fallback control. Appendix K.4 / Fig. 4 provides the worst-case qualitative picture under a deliberately out-of-scope non-monotone folding/saturation distortion. Under an always-accept policy, the induced subspace error rises sharply as drift strengthens. Under the contract gate, acceptance is suppressed and the system falls back to no-update mode rather than compounding the error through continued misaligned adaptation. The key point is not that the theorem extends beyond its assumptions, but that the deployment protocol remains well-behaved when the assumptions fail.
>
> ## 3. Heavy quantization, ties, and tie-aware operation (Addressing KQ3).
> Heavy quantization and sensor saturation are boundary conditions for any rank-based method, and the paper treats them explicitly. Theorem 3.1 shows that the guarantee degrades with the tie-dependent term $\delta_{tie}$, Appendix G.1 defines stable tie-breaking and logs $\text{TieMass}(v)$, and Proposition K.3 states the boundary clearly: once tie probability becomes sufficiently large, the induced rank map becomes intrinsically ill-posed and rank-based estimators incur irreducible ambiguity.
>
> Operationally, the protocol already does three things: (1) it controls numerical sensitivity through the clipped probit map in Eq. (7); (2) it uses stable tie handling and exposes TieMass directly in the logs; (3) it maps diagnostics to bounded remediation actions through Sec. 4.4 / Algorithm 2 before full refusal is triggered. When rank-related diagnostics dominate, the protocol can increase $\tau_{rank}$, adjust the number of slicing directions, strengthen ridge stabilization, or increase sketch size.
>
> If these are insufficient—and ties have become severe enough to destroy the macroscopic order required by the contract—then the auditing gate in Eq. (103) rejects operation rather than forcing an invalid alignment update. **While dynamically adjusting the clipping boundary $\alpha$ could serve as a secondary heuristic, the framework deliberately prioritizes structural safety over forced alignment. The resulting operational guarantee is highly robust**: the valid region of the contract is explicit, the surrogate in Eq. (27) is empirically calibrated, and once the heavy-tie boundary is crossed, the system safely refuses adaptation instead of failing silently.

---

> > ### Author Rebuttal · Reviewer_9nDy · 2026-04-02
> >
> > Thank you for your reply. My question has been answered, and I will keep my score.

---

> > > ### Author Response · Authors · 2026-04-04
> > >
> > > Thank you very much for your confirmation. We are glad that our rebuttal has fully answered your questions, and we sincerely appreciate your positive evaluation of our work.
> > >
> > > To reflect this constructive discussion more clearly, we will explicitly detail the per-deployment recalibration process, the fallback safety guarantees, and the heavy-tie boundary conditions in the final version.
> > >
> > > Thank you again for your thorough review and valuable support.

---

### Official Review · Reviewer_Q2o9 · 2026-03-10

**Soundness:** 3
**Presentation:** 4
**Significance:** 3
**Originality:** 4
**Overall Recommendation:** 4
**Confidence:** 2

**Summary:**

The paper proposes the Stable Spectral Copula Alignment (SSCA) method, which addresses the alignment failure caused by deployment shift in multimodal learning by constructing a comprehensive solution integrating Copula theory, optimal transport, and spectral learning.

**Compliance With Llm Reviewing Policy:**

Affirmed.

**Final Justification:**

After carefully reviewing both the paper and the authors’ rebuttal, I assign an overall score of **4 (Weak Accept)**. The rebuttal fully addressed all my main concerns regarding the work’s soundness, clarity, and experimental validation, which significantly improved my assessment. While the paper shows only incremental originality and modest significance, its methodology is solid and the contributions are sufficient for publication. I therefore maintain a balanced positive judgment and support a weak accept recommendation.

**Key Questions For Authors:**

To clarify, I am not an expert in this domain and can only raise relatively broad questions for the authors' consideration:
1. In real-world scenarios with a high proportion of non-monotonic drift, can the robustness be further enhanced by dynamically adjusting the weights of error proxies or integrating non-monotonic detection modules?
2. How will SSCA's performance change in complex scenarios involving partial modality missing (e.g., random missing of 1-2 modalities) combined with marginal distortion?

**Limitations:**

yes

**Strengths And Weaknesses:**

The paper excels in all four key dimensions: soundness, presentation, significance, and originality. It features rigorous theory, clear expression, impactful contributions, and notable innovation. As a non-expert in this field, I believe the overall approach of the paper is fundamentally sound, with sufficient research effort and innovation. While there is room for improvement in aspects such as computational efficiency and experimental design, the proposed SSCA method generally provides a practical solution for the field of robust multimodal alignment.

---

> ### Author Rebuttal · Authors · 2026-03-29
>
> # Response to Reviewer Q2o9
>
> Thank you for the thoughtful and constructive review. Your questions raise two practically important deployment scenarios beyond the paper’s core in-scope setting: high proportions of non-monotone drift, and partial modality missingness combined with marginal distortion. Below, we clarify how the current SSCA framework handles these regimes mechanistically.
>
> ## 1. Dynamic weighting or explicit non-monotone detection (Addressing KQ1).
> In the current paper, the coefficients in Eq. (27) are intentionally static and are calibrated from healthy unlabeled data through Eq. (28), after which the deployment decision is made by the gate in Eq. (29). This is a deliberate design choice rather than a modeling restriction. The contract prioritizes auditability: each term in the proxy bound corresponds to an observable quantity, the coefficients are obtained through a transparent label-free quantile-regression procedure, and the final deployment decision remains directly inspectable.
>
> At the same time, the framework is structurally compatible with more adaptive extensions. The diagnostic dictionary already separates rank-, coupling-, sampling-, and numerical-instability sources, and Sec. 4.4 already maps these diagnostics to bounded remediation actions under a compute budget. A non-monotonicity detector could therefore be introduced either as an additional proxy term or as a side channel that modulates remediation or fallback thresholds. In that sense, the framework is structurally primed for such dynamic reweighting extensions.
>
> **The framework’s handling of non-monotone drift deliberately prioritizes strict operational safety**. Once the rank-based invariance assumption is no longer trustworthy, the system should reduce acceptance and fall back rather than extrapolate the same guarantee beyond its assumptions. This is exactly the behavior shown in Appendix K.4 / Fig. 4: under a deliberately non-monotone folding/saturation distortion, gate acceptance drops as the violation strengthens, and the protocol avoids the error escalation seen under always-accept updates. Thus, the present contribution establishes an auditable foundation for safe refusal, upon which fully dynamic adaptation mechanisms can be naturally built.
>
> ## 2. Partial modality missingness combined with marginal distortion (Addressing KQ2).
> This is an important deployment scenario, and the current submission already defines the protocol needed to study it. Appendix I.5 specifies the missing-modality setting via Bernoulli masks and available-only fusion, and Appendix I.6 defines the Mask-Aware Standardized Imputation (SI) module. In addition, copula stabilization is applied per available modality before cross-modal fusion. From the current design, this extension is technically natural rather than requiring architectural redesign: surviving modalities are stabilized individually before entering fusion or standardized imputation, reducing the direct propagation of marginal distortion into downstream alignment.
>
> Equally importantly, the method does not require every modality to be present at every step. Available-only fusion is defined in Eq. (93), and the SI path in Eqs. (94)-(97) provides an imputation mechanism trained on the training split only. The same gate then monitors whether the current proxy budget and empirical eigengap still indicate a stable operating regime. In this sense, the existing framework already contains the ingredients needed for the combined missingness+distortion setting.
>
> **While a dedicated missing-modality theorem and the exact combinatorial stress-test fall outside the primary scope of the main paper, the framework natively accommodates this intersection**. Evaluating this intersection is a straightforward operational extension of the explicitly defined Appendix I.5-I.6 protocol, rather than requiring new architectural designs.
>
> Conceptually, the current contract relies on recoverable cross-modal dependence and an identifiable eigengap. If missingness becomes severe enough to fundamentally weaken the shared dependence structure, the diagnostics and eigengap indicate the contract no longer applies, and the auditing gate safely refuses adaptation. The intended answer is two-part: structural resilience while dependence remains identifiable, and safe refusal once that structure breaks down.
>
> **Framing as an auditable deployment protocol**: These questions help clarify the intended contribution of SSCA. The paper is best read as an auditable deployment protocol: its purpose is to determine when alignment can be trusted, when bounded remediation is justified, and when fallback is the correct deployment action.

---

> > ### Author Rebuttal · Reviewer_Q2o9 · 2026-04-03
> >
> > Thank you for your reply. My question has been fully resolved. I will keep my score to recommend accept.

---

> > > ### Author Response · Authors · 2026-04-04
> > >
> > > Thank you very much for your kind confirmation. We are delighted that our explanations have fully resolved your questions, and we sincerely appreciate your positive evaluation of the paper.
> > >
> > > To reflect this constructive exchange, we will explicitly incorporate the extended discussions on non-monotone drift adaptations and missing-modality protocols into the revision.
> > >
> > > Thank you once again for your careful evaluation and valuable support.

---

### Official Review · Reviewer_uGG8 · 2026-03-11

**Soundness:** 3
**Presentation:** 3
**Significance:** 3
**Originality:** 3
**Overall Recommendation:** 5
**Confidence:** 2

**Summary:**

Multimodal alignment models are vulnerable to deployment-time distribution shifts, particularly under strictly monotone marginal distortions such as normalization drift, sensor gain changes, and compression or quantization effects. Standard alignment objectives often degrade under such shifts because cross-modal dependence is entangled with marginal-sensitive geometric structure. Stable Spectral Copula Alignment (SSCA) introduces a framework that aims to isolate copula-invariant dependence while suppressing marginal distortions. A central technical component is an empirical Davis–Kahan–style inequality that bounds subspace deviation through observable proxy diagnostics. In contrast to classical perturbation bounds that rely on unobservable population quantities, the derived inequality connects subspace stability to measurable diagnostic proxies. Empirical evaluation on CMU-MOSEI, MELD, MSCOCO, and CC3M-500K benchmarks demonstrates consistent reductions in performance degradation under multiple distortion settings compared with established baselines.

**Compliance With Llm Reviewing Policy:**

Affirmed.

**Key Questions For Authors:**

1. Could additional quantitative evaluation of the Stability Gate be provided, particularly regarding its detection behavior under distribution shifts? For example, reporting false-positive and false-negative rates when the gate activates (or fails to activate) the Fallback Mode under out-of-scope or non-monotone perturbations (e.g., adversarial perturbations or pixel permutations) would help clarify the reliability of the mechanism.
2. To better assess the practical utility of SSCA, could a concrete empirical breakdown of the training and inference overhead be provided relative to standard contrastive pipelines such as InfoNCE or SigLIP? It would also be helpful to quantify the computational impact of the hard-coupling approximation.
3. How does the computational complexity of the diagonal-stabilized block-spectral learning module scale with the shared subspace dimensionality and the number of modalities? Empirical scaling results or complexity analysis would help clarify the method’s applicability to large multimodal systems.

**Limitations:**

yes

**Strengths And Weaknesses:**

Strengths:
1. The work presents a mathematical formulation for decomposing empirical perturbation errors, where each component is addressed through a three-module architecture. The derivation of the Davis–Kahan–style inequality connects a classical perturbation bound with observable diagnostic proxies. Experimental results on four datasets, together with ablation studies, provide empirical evidence regarding the contribution of the proposed components.
2. The presentation is generally well organized, and the relationship between the three modules (Copula Stabilization, Hub Coupling, and Spectral Learning) and the underlying error decomposition is clearly described.
3. The proposed "Stability Gate" aims to provide practical utility by enabling label-free reliability monitoring, which may help support safer deployment in practical scenarios.

Weaknesses:
1. The theoretical guarantees rely on the assumption of coordinate-wise monotone marginal distortions, which may not fully capture more complex forms of distribution shift encountered in real-world deployments. In addition, in edge cases such as extreme quantization or heavy score ties, the rank-based transformations may become unstable and potentially degrade alignment quality. Although these limitations are briefly acknowledged, the discussion of their practical implications and possible mitigation strategies remains limited.
2. Compared with relatively lightweight contrastive pipelines (e.g., InfoNCE or SigLIP), the proposed framework introduces additional algorithmic components. In large-scale pretraining or resource-constrained environments, these operations could introduce additional computational overhead. A more detailed discussion of scalability, would help clarify the practical feasibility of the approach.

---

> ### Author Rebuttal · Authors · 2026-03-29
>
> # Response to Reviewer uGG8
>
> Thank you for the thoughtful and supportive review. We especially appreciate your recognition of the paper’s error-source decomposition, the alignment between the three modules and the theory, and the practical relevance of the Stability Gate. Your questions focus on gate reliability, computational overhead, scaling, and behavior near the heavy-tie boundary.
>
> ## 1. Quantitative evaluation of the Stability Gate (Addressing KQ1).
> The paper already provides a three-part evidence chain for gate reliability:
>
> •	**Gradual Drift**: Table 4 shows that the combined diagnostic achieves AUROC 0.94, AUPRC 0.81, and an average lead time of 7 batches before substantial performance drop.
>
> •	**Mixed-Scope Stress Test**: Fig. 3 and Table 12 evaluate the gate under mixed-scope drift. Table 12 provides the operating-point numbers on LAION-1M mixed-scope drifts: FPR 3.2%, FNR 2.8%, and intervention accuracy 94.6%, achieving the best combined FPR/FNR profile compared to MMD, logit-entropy monitoring, and cosine-thresholding.
>
> •	**Policy-Level Outcomes**: Table 6 evaluates the gate as a deployment policy rather than as a detector alone: refusal remains moderate, Unsafe Accept is low, Regret is small, and Net Benefit is consistently positive.
>
> ## 2. Training / inference overhead and practical feasibility (Addressing W2 & KQ2).
> The key point is that the paper profiles the alignment stage under the **frozen-feature protocol** actually studied here, not end-to-end pretraining. Table 19 and Appendix H.3 make this explicit: all measurements exclude data loading and encoder feature extraction. Within that setting, the empirical profile is favorable rather than burdensome. At batch size 256, Table 19 reports:
>
> •	InfoNCE:              18.2k samples/s | 5.1 GB | 14.1 ms/batch
>
> •	SigLIP:                 18.5k samples/s | 5.1 GB | 13.8 ms/batch
>
> •	MMML:                19.2k samples/s | 4.8 GB | 13.3 ms/batch
>
> •	SSCA (SoftRank): 22.5k samples/s | 4.2 GB | 11.4 ms/batch
>
> •	SSCA (Sketch):     24.8k samples/s | 3.9 GB | 10.3 ms/batch
>
> The supported claim is therefore **precisely scoped and highly actionable**: within the alignment-stage regime studied in this paper, SSCA is computationally competitive and often cheaper than standard contrastive alternatives. Regarding the hard-coupling approximation, the released implementation uses it as the efficient default (Appendix G.6), while Appendix H reports the per-module asymptotic costs. Table 19 profiles the alignment stage under this default implementation.
>
> ## 3. Complexity scaling of the spectral module (Addressing KQ3).
> As formalized in Sec. 3.3 and Appendix H.1, the diagonal-stabilized block-spectral module uses implicit matrix-vector products, so the dominant spectral step scales as $\mathcal{O}(n_{iter} d^2 m k)$, where $d$ is the number of modalities, $m$ the batch size, $k$ the shared subspace dimension, and $n_{iter}$ the eigensolver iterations. The implementation therefore avoids explicit construction of the full block covariance matrix and the corresponding memory blow-up. Appendix H reports the asymptotic costs of stabilization, dependence estimation, hub coupling, and the spectral step separately.
>
> ## 4. Heavy quantization, ties, and boundary conditions (Addressing W1).
> Extreme quantization and heavy ties are genuine boundary conditions treated explicitly. Theorem 3.1 includes a tie-dependent instability term; Appendix G.1 specifies the stable tie-handling procedure and logs TieMass; and Proposition K.3 states the boundary clearly: once tie probability becomes large, the induced rank map becomes ill-posed and rank-based estimators incur irreducible ambiguity.
>
> Operationally, the intended behavior is not to force continued adaptation under an unsupported invariance assumption. Instead, Sec. 4.4 / Algorithm 2 provide bounded remediation actions before refusal, including increasing $\tau_{rank}$, expanding the number of slicing directions $S$, adjusting ridge stabilization $\lambda_{stab}$, or enlarging sketch size. If these are insufficient to restore a stable regime, the gate rejects operation and reverts to the corresponding no-update control. The same logic applies to non-monotone shifts: as shown in Appendix K.4 / Fig. 4, gate acceptance decreases as the violation strengthens, and the system falls back rather than compounding the error through unsafe continued adaptation. In this sense, the paper’s out-of-scope claim is deliberately operational: safe behavior through diagnosis, bounded remediation, and refusal when the contract no longer holds.

---

> > ### Author Rebuttal · Reviewer_uGG8 · 2026-04-01
> >
> > Thank you for the detailed response. This addresses my concerns, and I will keep my score to recommend accept.

---

> > > ### Author Response · Authors · 2026-04-04
> > >
> > > Thank you very much for the follow-up. We are glad that our response has fully resolved your concerns, and we sincerely appreciate your positive evaluation of the paper.
> > >
> > > Building on this discussion, we will ensure that the quantitative evaluations of the Stability Gate, computational profiling, and tie-handling boundaries are highlighted more clearly in the revision.
> > >
> > > Thank you again for your time, insightful questions, and support.

---

### Official Review · Reviewer_VkDP · 2026-03-13

**Soundness:** 3
**Presentation:** 2
**Significance:** 3
**Originality:** 3
**Overall Recommendation:** 4
**Confidence:** 3

**Summary:**

This paper proposes Stable Spectral Copula Alignment (SSCA), a multimodal alignment framework aimed at improving robustness under deployment shift. The central challenge that the authors address is that many multimodal systems are sensitive to distribution shifts that affect the feature marginals even when the underlying cross-modal dependence remains invariant. The key idea is that data order matters more than the magnitudes, and by constructing a copula based on the rank/quantile of the datapoints, the downstream decisions are less sensitive to sheer magnitude, which in turns allows the authors to learn correspondence/alignment across the modalities. The paper also discusses diagnostics methods that decide whether the model is in a “stable” regime, should undergo remediation, or should fall back conservatively. The reported results show that SSCA consistently reduces degradation relative to a broad set of baselines, including contrastive, correlation-based, transport-based, and recent robust multimodal methods.

**Compliance With Llm Reviewing Policy:**

Affirmed.

**Key Questions For Authors:**

1. How much of the improvement comes from copula stabilization alone versus the full pipeline?  The paper includes relevant baselines, but I would like a clearer discussion of when the hub coupling and spectral components are essential rather than merely beneficial.

2. What happens under shifts that are only partially monotone? Although  the paper clearly states that such cases are out of scope, still some empirical gains seem to extend beyond the formal contract. Initially, this seemed impressive, but I am wondering if the contract really corresponds to the reasons why the method shows improvement. Ideally, a synthetic experiment in which assumptions are violated and performance decays would be useful to justify the assumptions. It would generally help to clarify which observed improvements are theoretically motivated versus purely empirical.

**Limitations:**

yes

**Strengths And Weaknesses:**

Strengths

1. Interesting problem formulation.  The paper tackles an important robustness issue in multimodal learning: performance degradation under deployment-time distribution shift, especially when standard alignment methods are marginal-sensitive.
2. Coherent design.  The three modules are motivated by a common decomposition of error sources, and the paper presents them as parts of a single operational protocol.
3. Strong empirical results.  The experiments are broad and generally convincing. SSCA performs well on both classification and retrieval settings, and the gains under perturbation appear consistent across datasets and drift types.
4. Good ablation and baseline coverage.  I appreciated that the paper includes stronger targeted baselines such as “copula-only + strong objective” and pairwise OT variants, which helps support the claim that the full method adds value beyond any one component.

Weaknesses

1. Scope of guarantees is fairly narrow.  The strongest justification appears to hold primarily for approximately coordinate-wise monotone marginal distortions. This is a meaningful setting, but narrower than the broader deployment-robustness framing might initially imply. This must be received more as a presentation comment; I believe the paper might be over-promising on the theoretical component.
2. Limited clarity on failure modes outside the contract.  The paper does acknowledge out-of-scope shifts and advocates fallback, which is good, but I would have liked a sharper empirical characterization of exactly when the gate fails, when it becomes overly conservative, and how much performance is lost due to fallback.
3. Presentation could be cleaner.  The paper is ambitious and sometimes dense. In particular, the relation between the theoretical bound, the diagnostic proxies, and the calibrated decision rule could be explained more directly.

---

> ### Author Rebuttal · Authors · 2026-03-29
>
> # Response to Reviewer VkDP
>
> Thank you for the thoughtful and constructive review. We especially appreciate your recognition of the problem formulation, the coherent protocol design, the broad empirical study, and the targeted baselines. Your comments focus on exactly the places where a deployment paper must be most precise: the scope of the formal contract, the contribution of each module beyond copula stabilization alone, and the behavior of the system once that contract no longer holds.
> ## 1. Scope of guarantees and framing (Addressing W1).
> SSCA establishes a rigorous, explicitly scoped guarantee rather than claiming universal robustness to arbitrary deployment shifts. Its strongest formal guarantee is explicitly scoped to approximately coordinate-wise monotone marginal distortions in feature space, as defined in Sec. 2.2 and formalized through Theorem 3.1 and Theorem 4.1. The actionable inequality in Eq.(27), together with the calibration and gate in Eqs.(28)-(29), supports an auditable stability contract: in-scope drifts admit bounded subspace-perturbation certificates and calibrated acceptance rules, while out-of-scope drifts trigger a conservative gate/fallback path. We agree that this boundary should be stated even more explicitly in the Abstract and Introduction, and we will revise the framing accordingly.
> ## 2. Copula stabilization alone vs. the full pipeline (Addressing KQ1).
> This is a central question, and the paper already contains two complementary answers. First, the critical baseline comparison in Sec. 5.3 / Appendix Table 20 shows that Copula-only + SigLIP is indeed strong, but full SSCA still improves it materially: on CMU-MOSEI, degradation decreases from 2.3 to 1.8; on MELD, from 2.2 to 1.5. Second, the ablation study in Table 7 is even more direct: removing hub coupling increases degradation to 3.8/3.5, and replacing the spectral step with a contrastive objective increases it to 3.5/3.2. Together, these results show that copula stabilization is necessary but not sufficient. It addresses marginal sensitivity, while the remaining gain is consistently explained by multiway coherence (hub coupling) and shared-subspace identifiability under sampling/numerical noise (block-spectral learning). We will make this distinction more explicit in Sec. 5.3.
> ## 3. Partially monotone / out-of-contract shifts (Addressing W2 & KQ2).
> We appreciate the opportunity to further sharpen the boundary between theoretical guarantees and empirical robustness. The controlled monotone experiments in Table 1 are the results most directly backed by the contract. The realistic deployment drifts in Table 3 are empirical robustness results under the same protocol, serving as practical complements to the formal guarantees. Appendix K.4/Fig. 4 then shows what happens when the contract is explicitly violated: a deliberately non-monotone folding/saturation distortion breaks rank preservation; under an always-accept policy, induced subspace error escalates; under SSCA, gate acceptance decreases as the violation strengthens, and the system reverts to the corresponding safe no-update mode. The point of this experiment is therefore not to suggest that the theorem extends beyond its assumptions, but to show that once the contract fails, the protocol fails safely rather than silently.
> ## 4. Gate conservativeness and fallback penalties (Addressing W2).
> The existing evidence chain already answers this quantitatively. Table 4 shows AUROC 0.94, AUPRC 0.81, and a mean lead time of 7 batches under gradual drift. Table 6 shows moderate refusal rates (12.5%-18.9%), low unsafe-accept rates (1.7%-2.6%), small regret (0.3-0.5), and consistently positive net benefit. Appendix Table 12 reports FPR 3.2%, FNR 2.8%, and intervention accuracy 94.6% on LAION-1M mixed-scope drifts. Appendix K.3 further makes the accounting explicit: REFUSE, UNSAFEACCEPT, REGRET, and BENEFIT are all measured against the compute-matched fallback under the same drift realization and evaluation seed. Together, these results support the intended interpretation that the gate is neither decorative nor excessively conservative. When the gate rejects, the correct interpretation is a policy-level safe-failure claim, not an out-of-scope accuracy theorem.
> ## 5. Presentation clarity (Addressing W3).
> We agree that the exposition can be made more direct. In the revision, we will present the theory-to-deployment chain in one pass: (i) the perturbation decomposition in Eq.(1); (ii) the actionable proxy inequality in Eq.(27); (iii) the calibration and thresholding steps in Eqs.(28)-(29); and (iv) Tables 4-6 plus Appendix K, which show how the protocol drives remediation and fallback in practice. We believe this will make the contract boundary, the empirical boundary, and the gate/fallback semantics much clearer.
>
> We thank you again for the careful review. Your comments help clarify the paper’s scope, module roles, and out-of-contract behavior without changing technical claims.

---

> > ### Author Rebuttal · Reviewer_VkDP · 2026-04-04
> >
> > I thank the authors for their detailed replies. My concerns have been mostly addressed.

---

> > > ### Author Response · Authors · 2026-04-05
> > >
> > > Thank you for the clarification and careful follow-up. We are glad that our response has addressed your concerns.
> > >
> > > We will make three points more explicit in the revision: the monotone-marginal scope of the formal contract, the distinct roles of copula stabilization vs. hub/spectral components, and the operational interpretation of out-of-contract behavior.
> > >
> > > We are grateful that your review helped sharpen the paper’s presentation and overall framing.

---

### Decision · Program_Chairs · 2026-04-30

**Decision:**

Accept (regular)

**Comment:**

The paper proposes Stable Spectral Copula Alignment (SSCA), a deployment-oriented framework for robust multimodal alignment under monotone marginal shifts. The problem is important, and the use of ideas from copula theory to isolate dependence structure is both novel and technically meaningful. Reviewers consistently found the overall framework strong and the empirical evaluation convincing. In particular, the integration of copula stabilization, hub coupling, spectral learning, and an auditable stability gate was viewed as practically relevant. The reviews are uniformly positive after discussion, and I therefore support acceptance.